# Astrocytes refine cortical connectivity at dendritic spines

W Christopher Risher[1,2], Sagar Patel[1], Il Hwan Kim[1], Akiyoshi Uezu[1], Srishti Bhagat[2], Daniel K Wilton[3], Louis-Jan Pilaz[4], Jonnathan Singh Alvarado[1], Osman Y Calhan[1], Debra L Silver[1,2,4,5], Beth Stevens[3], Nicole Calakos[2,5,6], Scott H Soderling[1,2,5], Cagla Eroglu[1,2,5]*

[1]Department of Cell Biology, Duke University Medical Center, Durham, United States; [2]Department of Neurobiology, Duke University Medical Center, Durham, United States; [3]Department of Neurology, FM Kirby Neurobiology Center, Boston Children's Hospital, Harvard Medical School, Boston, United States; [4]Department of Molecular Genetics and Microbiology, Duke University Medical Center, Durham, United States; [5]Duke Institute for Brain Sciences, Durham, United States; [6]Department of Neurology, Duke University Medical Center, Durham, United States

**Abstract** During cortical synaptic development, thalamic axons must establish synaptic connections despite the presence of the more abundant intracortical projections. How thalamocortical synapses are formed and maintained in this competitive environment is unknown. Here, we show that astrocyte-secreted protein hevin is required for normal thalamocortical synaptic connectivity in the mouse cortex. Absence of hevin results in a profound, long-lasting reduction in thalamocortical synapses accompanied by a transient increase in intracortical excitatory connections. Three-dimensional reconstructions of cortical neurons from serial section electron microscopy (ssEM) revealed that, during early postnatal development, dendritic spines often receive multiple excitatory inputs. Immuno-EM and confocal analyses revealed that majority of the spines with multiple excitatory contacts (SMECs) receive simultaneous thalamic and cortical inputs. Proportion of SMECs diminishes as the brain develops, but SMECs remain abundant in Hevin-null mice. These findings reveal that, through secretion of hevin, astrocytes control an important developmental synaptic refinement process at dendritic spines.

*For correspondence: c.eroglu@cellbio.duke.edu

**Competing interests:** The authors declare that no competing interests exist.

**Reviewing editor**: Liqun Luo, Howard Hughes Medical Institute, Stanford University, United States

## Introduction

The cerebral cortex receives synaptic inputs from various cortical and subcortical areas including the thalamus. In the mouse brain, innervation of the cortex by projecting neurites called axons begins during embryonic development and continues for the first several postnatal days (*Garel and Lopez-Bendito, 2014*). Only after the axons project to their approximate target areas, hosting their suitable postsynaptic partners, does an intense period of synapse formation occur, corresponding roughly to the second and third postnatal weeks in mice (*Li et al., 2010*). Cortical excitatory synapses, which primarily use the neurotransmitter glutamate, are formed between dendritic protrusions called spines and axonal projections coming from two predominant inputs: intracortical and thalamic. Though the bulk of the cortical synapses from both of these inputs are made during the same early postnatal synaptogenic period (P5–P21) (*Nakamura et al., 2005*), whether they form through similar or differential mechanisms is unclear.

Intracortical and thalamocortical connections can be distinguished as they primarily contain either vesicular glutamate transporter-1 (VGlut1) or VGlut2 in their presynaptic terminals, respectively (*Kaneko and Fujiyama, 2002*). In most cortical areas, VGlut1-positive (VGlut1+) intracortical projections greatly outnumber the VGlut2+ thalamic projections. The cellular and molecular mechanisms through which

**eLife digest** The central nervous system—which is made up of the brain and spinal cord—processes information from all over the body. The information travels through cells called neurons, which connect to each other at junctions called synapses. A single neuron can receive information from many different places because it is covered with protrusions known as dendritic spines that enable it to form synapses with a variety of other neurons.

In recent years, it has become apparent that brain cells other than neurons can influence synapse formation. The most abundant cells in the central nervous system are star-shaped cells known as astrocytes, which secrete molecules that control the timing and extent of synapse formation. Many previous studies on synapses have used a type of neuron found in the eye—called retinal ganglion cells—because these cells can be purified and grown in the laboratory in the absence of astrocytes. Under these conditions, they form very few synapses. However, in the presence of astrocytes the retinal ganglion cells form many more synapses, which is thought to be due to a protein called hevin and several other proteins that are secreted by the astrocytes.

Risher et al. studied a region of the brain called the cerebral cortex in mice that were missing hevin. In the cortex of normal mice, the neurons generally form synapses with other neurons within the cortex, or with neurons from other parts of the brain that send long-distance projections into the cortex. The experiments revealed that fewer of these long-distance synapses formed in the cortex of the mice missing hevin compared to normal mice. When hevin was injected directly into the brains of the mice, more long-distance synapses were formed.

Using a technique called three-dimensional electron microscopy, Risher et al. examined the structure of the synapses. In mice missing hevin, the synapses were much smaller and the dendritic spines were thin and long, indicating that they were not fully grown. The images also show that in normal mice, the dendritic spines often have multiple synapses when the animal is young, but many are lost as the brain matures so that only a single synapse remains in each dendritic spine. However, multiple synapses persist in the dendritic spines of mice lacking hevin, which could lead to competition between short and long distance synapses and may contribute to neurological diseases.

These results indicate that astrocytes are crucial for controlling the formation of synapses in dendritic spines. In humans, defects in hevin have been implicated in autism, schizophrenia and other neurological conditions. Future studies will seek to determine the precise role of astrocytes in these conditions, which may help us to develop new therapies.

thalamocortical connections are established and maintained, despite steep competition from the vastly more abundant intracortical axons, have yet to be elucidated.

Concurrently with the synaptogenic period, non-neuronal cells called astrocytes begin to populate the cortex, producing and secreting factors that promote synaptogenesis (*Eroglu and Barres, 2010*). For example, hevin (a.k.a. synaptic cleft-1 or SPARC-like 1) is an astrocyte-secreted extracellular matrix protein that localizes to the clefts of excitatory synapses (*Johnston et al., 1990*; *Lively et al., 2007*) and promotes excitatory synaptogenesis (*Kucukdereli et al., 2011*). Using a combination of in vivo and in vitro approaches, here we show that in the cortex, hevin is specifically required for the formation of thalamocortical synapses. Moreover, using three-dimensional reconstructions of serial-section electron microscopy (ssEM)-imaged dendrites and axons, we show that P25 hevin KO dendritic spines often make multiple excitatory contacts with different axons.

Multiply-innervated spines were identified nearly a half-century ago by *Jones and Powell (1969)*. The majority of these spines make two distinct types of synapses: one excitatory with an asymmetric postsynaptic density (PSD) opposed to an axon with round presynaptic vesicles, and one inhibitory synapse with a symmetrical PSD and flattened vesicles (*Jones and Powell, 1969*; *Knott et al., 2002*; *Chen et al., 2012*). A small percentage, however, make multiple excitatory contacts (termed SMECs, i.e., spines with multiple excitatory contacts). We found that SMEC structures are frequent earlier in development at P14 in WT but essentially disappear by P25, indicating that they represent a transient stage in synaptic spine maturation. This developmental refinement is impaired in hevin KO mice. Moreover, using both confocal imaging as well as immunolabeling of electron micrographs with antibodies specific to VGlut1 and VGlut2, we found that these SMECs often contact thalamic and cortical

axons simultaneously. These results suggest that, during cortical synaptic development, dendritic spines serve as sites of competition between thalamic and cortical axons. Through secretion of hevin, astrocytes help maintain thalamic inputs onto cortical neurons and facilitate resolution of SMECs into singly-innervated spines.

## Results

### Loss of hevin results in deficient thalamocortical synaptic connectivity

The astrocyte-secreted synaptogenic protein hevin (a.k.a. SPARC-like 1/SPARCL-1 or Synaptic Cleft-1/SC1) increases the number of synapses made between retinal ganglion cells (RGCs) in vitro and is required for the correct formation and maturation of RGC synapses onto their postsynaptic target, the superior colliculus (*Kucukdereli et al., 2011*). Hevin expression is not restricted to the retinocollicular system. Hevin is expressed throughout the cortex in a developmentally regulated manner, peaking during P15–P25, a time period that coincides with intense synapse formation, maturation and elimination events in the cortex (*Figure 1A,B*). Staining for cell-specific markers confirms that hevin expression is largely restricted to astrocytes in the cortex (*Figure 1C–E*).

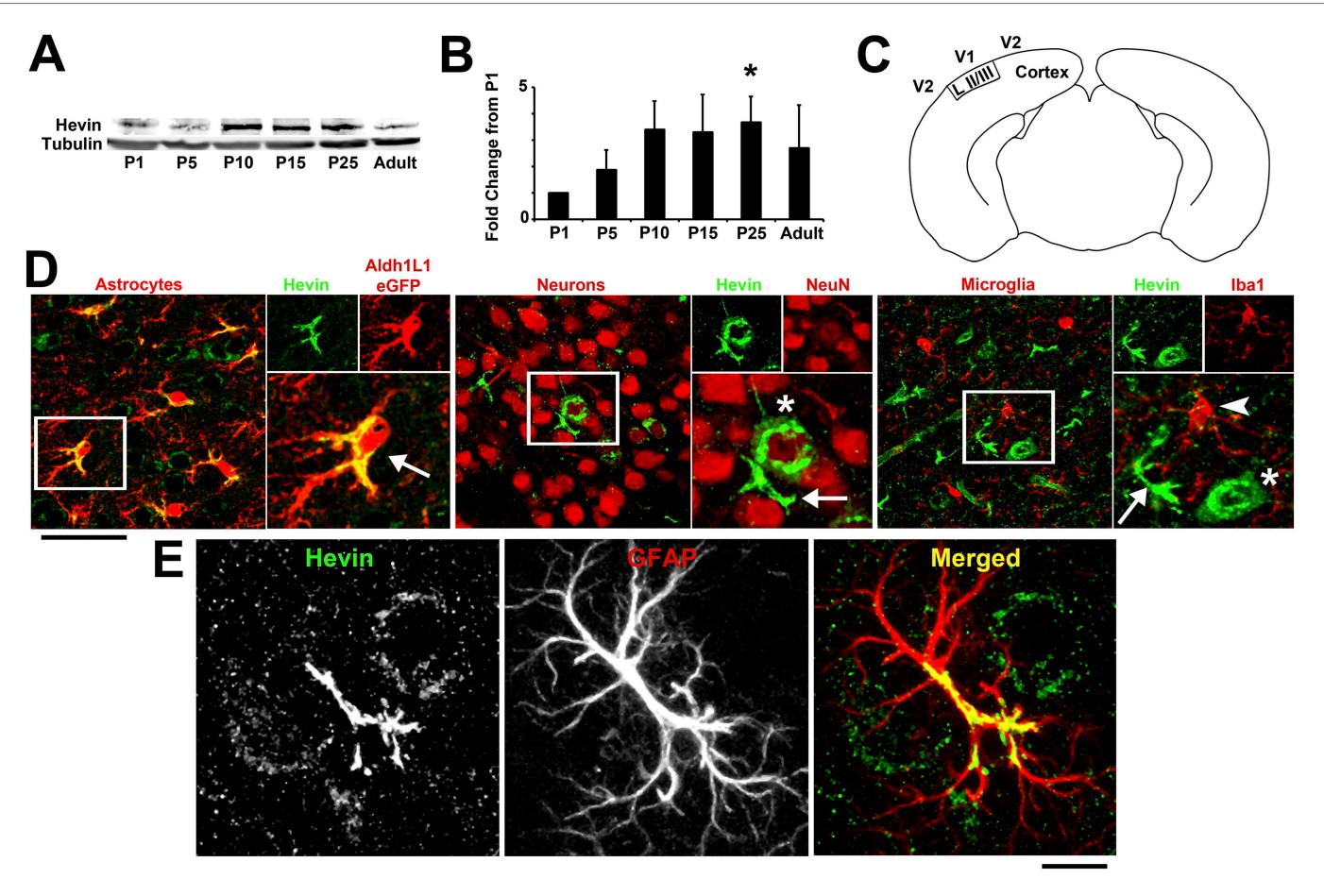

**Figure 1**. Hevin expression by astrocytes is developmentally regulated in the cortex. (**A**) Representative Western blots showing the developmental timeline for hevin expression in mouse cortex and hippocampus (tubulin was used as a loading control). (**B**) Quantification of Western blot analysis of hevin expression shows high expression between P15–P25. Data is presented as fold change compared to P1 levels (n = 3 animals per age; p < 0.05; one-way ANOVA with Dunnett's post hoc test). (**C**) Schematic diagram of a coronal slice through mouse brain shows the synaptic zone of primary visual cortex (V1) where EM, IHC and Golgi-cox staining analyses were performed. Layer II/III neurons of the visual cortex heavily project their dendrites to this region (**D**) IHC staining reveals that hevin expression (green) overlaps with all astrocytes (left, arrow) and a small subset of neurons (middle, asterisk) in V1, with no overlap seen with microglia (right, arrowhead). Cell-specific markers in red: Aldh1L1-EGFP for astrocytes, NeuN for neurons, Iba1 for microglia. Scale bar, 50 µm. (**E**) The rarely occurring GFAP+ astrocytes (red) in healthy visual cortex also express hevin (green). Scale bar, 10 µm.

In order to determine the role of hevin in cortical synaptic development, we investigated synaptic connectivity in the synaptic zone (S/Z, a.k.a. Layer I) of the mouse primary visual cortex (V1 region) in littermate age-matched hevin KO and wild-type (WT) mice using immunohistochemistry (IHC). Excitatory pyramidal neurons from upper cortical layers project extensive dendritic trees to this region and form a large number of the cortical synaptic connections. Cortical neurons receive two main classes of excitatory inputs: (1) the intracortical connections that are VGlut1 positive (VGlut1+), and (2) the sensory pathway inputs from the thalamus that are VGlut2+. The majority of the excitatory connections within the S/Z are of intracortical origin. On the other hand, the bulk of thalamocortical connections are made onto layer IV with a subset projecting to other layers including S/Z (*Figure 2A*) (*Kaneko and Fujiyama, 2002*; *Khan et al., 2011*). To determine the role of hevin in the formation of these different classes of cortical synapses, we quantified the number of synaptic puncta as the co-localization of the presynaptic VGlut1 or VGlut2 with postsynaptic PSD95 at P25 in littermate hevin-KO and WT mice. This synapse quantification assay takes advantage of the fact that pre- and post-synaptic markers are within separate compartments (axons and dendrites, respectively), but they appear to co-localize at synapses due to their close proximity (*Ippolito and Eroglu, 2010*). Surprisingly, we found that the number of VGlut1+ intracortical synapses in the S/Z of hevin KOs were significantly higher when compared to littermate WTs at P25 (*Figure 2B*). In contrast, thalamocortical VGlut2+ synapses were profoundly reduced in hevin KOs compared to WTs at P25 (*Figure 2C*). It is important to note that the appearance of co-localized VGlut/PSD95 puncta is not merely due to chance, since the randomization of puncta by rotating the channels out of alignment

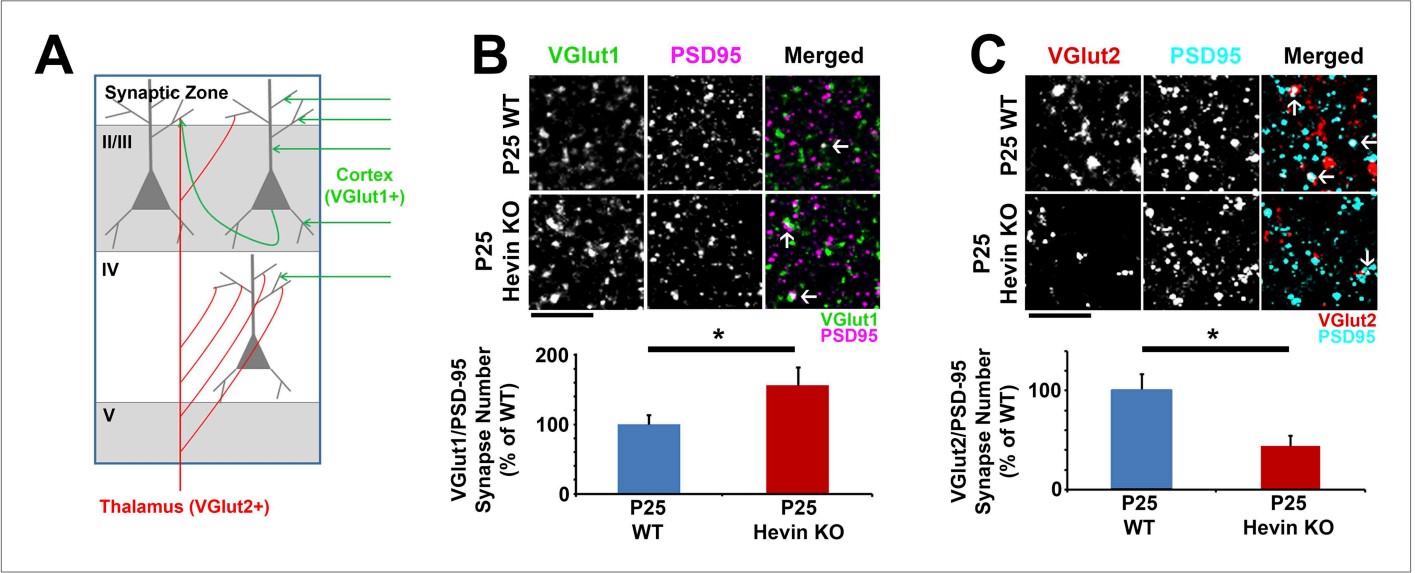

**Figure 2**. Hevin is required for proper thalamocortical innervation of V1. (**A**) Schematic of synaptic input to area V1. Most inputs are intracortical (green) and contain primarily VGlut1. The thalamus sends VGlut2+ projections (red) to various layers of V1 (primarily Layer IV). Competition occurs between VGlut1+ and VGlut2+ terminals for the same postsynaptic targets. (**B**) Co-localization of VGlut1 (green) and PSD95 (magenta) revealed an increase in intracortical synapses (co-localized puncta; arrows) in the synaptic zone of P25 hevin KO V1 (n = 3 z-stacks per animal, 5 animals per genotype; p < 0.05, nested ANOVA). Scale bar, 5 µm. (**C**) VGlut2 (red) and PSD95 (cyan) co-localization (arrows), representing thalamocortical synapses, was significantly decreased in P25 hevin KO compared to P25 WT (n = 3 z-stacks per animal, 4 animals per genotype; p < 0.05, nested ANOVA). Scale bar, 5 µm.

The following figure supplements are available for figure 2:

**Figure supplement 1**. Co-localized synaptic puncta are not due to random chance resulting from dense synaptic staining.

**Figure supplement 2**. Deficient thalamocortical connectivity in hevin KO is not due to decreased cortical neuron density.

**Figure supplement 3**. Reduced thalamocortical synapse density in hevin KO V1 is not due to deficient geniculocortical connectivity.

**Figure supplement 4**. Electrophysiological analysis of V1 cortical neurons in hevin KO.

by 90° nearly abolished occurrence of co-localized puncta in both the WT and KO (*Figure 2—figure supplement 1*).

The severe loss of thalamocortical synapses in hevin KOs was not due to a gross cellular defect in hevin KO cortices, because we found that neuronal density and layering was similar between P25 WT and hevin KO (*Figure 2—figure supplement 2A*). However, we observed a significant reduction in VGlut2+ synaptic terminals across multiple cortical layers in the hevin KOs (*Figure 2—figure supplement 2B*). This reduction in thalamic synaptic terminals was not due to the lack of thalamic neurons in hevin KOs at the dorsal lateral geniculate nucleus (dLGN) that project to V1 (*Figure 2—figure supplement 3A*). Moreover, by using a viral approach to trace the thalamic axons that project to V1 cortex, we found that lack of hevin did not impair the ability of thalamic projections to reach the S/Z (*Figure 2—figure supplement 3B–D*).

Hevin expression in the cortex starts to increase by the end of first postnatal week (corresponding to the start of synaptogenic period), peaking in abundance during the critical periods of plasticity in the cortex; hevin levels also remain high in the adult (*Figure 1B*). Therefore, we next investigated whether loss of hevin also alters synapse numbers in early postnatal (P7) or adult (12-week-old) mice. The number of VGlut1+ synapses in P7 hevin KOs trended towards an increase when compared to littermate WTs, but this increase was not yet significant (*Figure 3B*). However, similar to P25, P7 hevin KO cortex showed a severe deficit in VGlut2+ synapses (*Figure 3C*). This finding indicates that hevin is required for the formation of thalamocortical synapses in the early developing mouse brain.

The specific loss of thalamocortical synaptic connectivity in hevin KOs was not merely a developmental delay in thalamocortical synaptogenesis, since VGlut2+ synapse density was still significantly lower in the 12-week old hevin KO adults compared to WT (*Figure 3E*). However, the number of VGlut1+ synapses in the hevin KOs returned to normal levels in the adult (*Figure 3D*), indicating that the changes we observed in the VGlut1+ synaptic connectivity in the developing (P25) hevin KOs are due to a transient offsetting of the reduced VGlut2+ connections by VGlut1+ intracortical synapses. In agreement with this possibility, recordings from P23–P26 mice did not show any significant differences in miniature excitatory postsynaptic currents (mEPSCs) in layer 2/3 pyramidal neurons of hevin KOs when compared to their littermate WTs (*Figure 2—figure supplement 4*).

Taken together, these results show that hevin is required for normal formation and maintenance of VGlut2+ thalamocortical connections in the cortex. Our findings also demonstrate that lack of hevin results in a transient increase in intracortical synapses. This increase in intracortical synapses could either be mediated through a homeostatic mechanism that compensates for lost thalamic input and/or be driven by a transient competitive advantage for cortical axons over the thalamic inputs to establish synapses.

## Hevin is sufficient to induce thalamocortical synapse formation

Our analyses of excitatory synaptic development in hevin KO mice revealed an important role for hevin in the development of thalamocortical circuitry (*Figures 2–3*). Therefore, we next investigated whether hevin is sufficient to induce thalamocortical synaptogenesis. To address this question we first utilized in vitro assays with purified cortical and thalamic neurons. To do so we immunopurified cortical neurons from hevin KO pups and plated them either alone or in the presence of equal number of purified thalamic neurons (*Figure 4A*). Next we treated these cells with growth media with or without hevin and determined the effect of hevin treatment on the number of VGlut1+ or VGlut2+ excitatory synapses made onto cortical neurons (*Figure 4B–E*). Hevin treatment did not increase the number of VGlut1+ synaptic puncta (determined as the co-localization of VGlut1 and PSD95) in either the cortical neuron only cultures or in the cortical/thalamic neuron co-cultures (*Figure 4B,C*). Hevin also did not significantly affect VGlut2/PSD95-positive synapses in the cortical neuron-only cultures (*Figure 4D,E*), which were already at low quantities due to the fact that only a small portion of cortical axons expresses VGlut2 (*Wallen-Mackenzie et al., 2009*). However, hevin treatment significantly increased VGlut2+ synapse formation onto cortical neurons in cortical-thalamic co-cultures (*Figure 4D,E*). These in vitro evidence strongly suggest that hevin specifically induces formation of thalamocortical synapses.

Virally tracing the thalamocortical axons that innervate the S/Z in hevin KOs revealed that thalamocortical projections are intact in the hevin KO (*Figure 2—figure supplement 3B–D*), but they do have defects in establishing thalamocortical synapses. Therefore, we next tested whether injection of hevin into the developing cortex was sufficient to increase thalamocortical synaptic connectivity in vivo. To do so, pure hevin protein was directly injected into Layer II/III of P13 hevin KO V1 (*Figure 4F*).

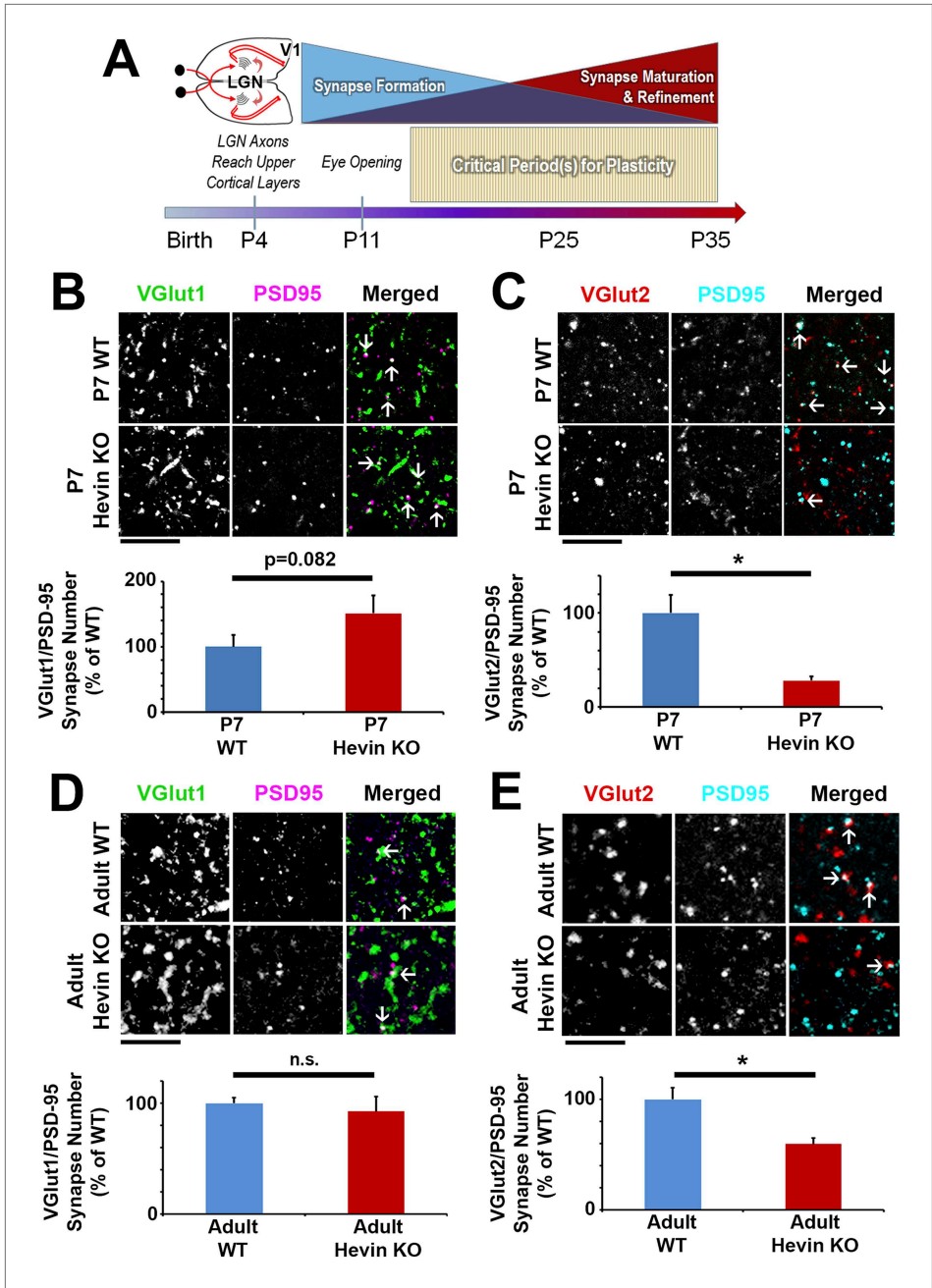

**Figure 3**. Hevin is important for both the formation and long-term maintenance of thalamocortical synapses. (**A**) Timeline of cortical synaptic development in mice. Axonal projections from the LGN reach their target areas in V1 shortly after birth. Around the time of eye opening, there is a period of intense synapse formation that gradually gives way to processes involved in synapse maturation and refinement, including synapse elimination. Multiple critical periods for different forms of plasticity in the visual system occur during this period of synapse formation and refinement. (**B**) At P7, the beginning of the synaptogenic period, co-localization of VGlut1 (green) and PSD95 (magenta) revealed a trend towards an increase in intracortical synapses (co-localized puncta; arrows) in the synaptic zone of hevin KO V1 (n = 3 z-stacks per animal, 3 animals per genotype; p = 0.082, nested ANOVA). (**C**) VGlut2 (red) and PSD95 (cyan) co-localization (arrows), representing thalamocortical synapses, was significantly decreased in P7 hevin KO compared to P7 WT (n = 3 z-stacks per animal, 3 animals per genotype; p < 0.05, nested ANOVA). Scale bars, 5 μm. (**D**) In the mature brain (12-weeks-old), hevin KO mice no longer have a discrepancy in VGlut1/PSD95 synaptic puncta when compared to WT (n = 3 z-stacks per animal, 5 animals per genotype; p > 0.05, Student's t test). (**E**) Deficient VGlut2/PSD95 synapse formation is still present in the mature hevin KO brains (n = 3 z-stacks per animal, 5 animals per genotype; p < 0.01, Student's t test). Scale bars, 5 μm.

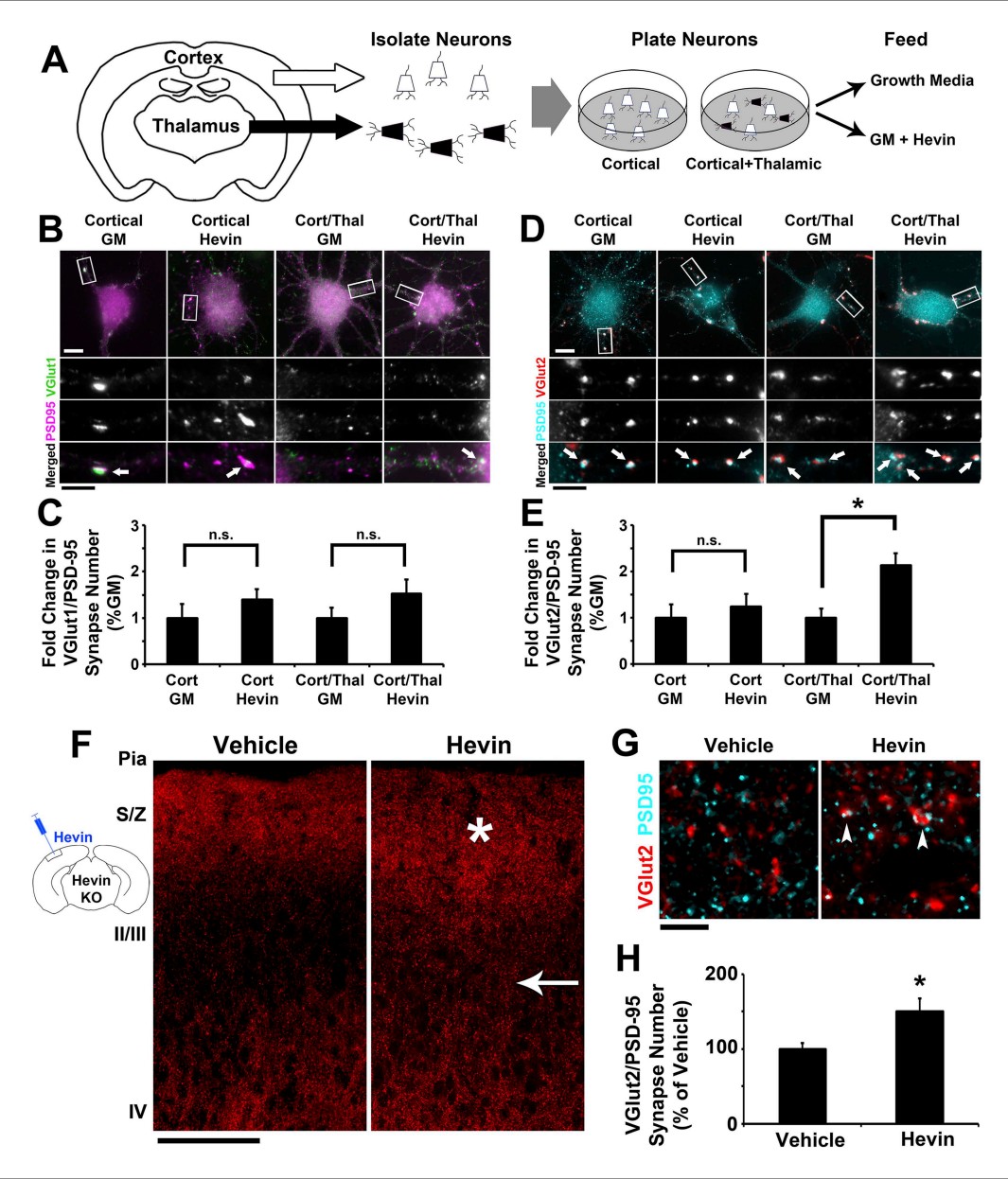

**Figure 4**. Hevin specifically induces thalamocortical synapse formation in vitro and in vivo. (**A**) Schematic of the cortical/thalamic neuron co-culture system. (**B**) Representative images of cortical neurons cultured for 14 days in vitro (DIV), either alone or in equal densities with thalamic neurons, with or without hevin treatment. Insets show individual channels for VGlut1 (green) and PSD95 (magenta) staining, as well as the merged image. Co-localized puncta (white, arrows) represent synapses. Scale bars: 10 μm (main image), 5 μm (inset). (**C**) Hevin did not induce VGlut1 synapse formation onto cortical neurons, with or without thalamic neurons also present (n = 30 cells per condition; p > 0.05, Student's *t* test between GM and hevin treatments). (**D**) Same as B, except VGlut2 appears as red and PSD95 in cyan. (**E**) Hevin strongly induces VGlut2/PSD95 synapse formation when cortical neurons are cultured together with thalamic neurons (n = 30 cells per condition; *p > 0.01, Student's *t* test between GM and hevin treatments). Upon hevin treatment VGlut2/PSD95 synapses are recruited heavily to neuronal soma and proximal dendrites. (**F**) Hevin protein was stereotactically injected directly into Layer II/III of hevin KO V1. When compared to vehicle-injected control, hevin-injected cortex had distinctly thickened VGlut2 staining throughout S/Z and upper II/III (asterisk), as well as a dense appearance of VGlut2+ axon tracks throughout II/III (arrow). Scale bar, 100 μm. (**G**) Hevin-injected V1 had more VGlut2/PSD95 co-localized puncta (arrowheads)
*Figure 4. Continued on next page*

*Figure 4. Continued*

than vehicle-injected controls. Scale bar, 5 µm. (**H**) Hevin injection significantly increased the number of VGlut2/PSD95 synapses in hevin KO cortex (n = 2 z-stacks per animal, 3 animals per treatment; p < 0.01, Student's *t* test).

The following figure supplement is available for figure 4:

**Figure supplement 1**. Hevin does not induce intracortical synapse formation in vivo.

After 3 days, the brains were fixed and immuno-stained for pre- and postsynaptic markers and then imaged by confocal microscopy approximately 100 µm laterally to the injection site (*Figure 4F*). Compared to vehicle-injected littermate controls, hevin-injected cortices showed a robust increase in VGlut2+ thalamocortical terminal staining throughout the S/Z and Layer II/III directly adjacent to the site of injection (*Figure 4F*, asterisk and arrow, respectively). Quantitative analysis of co-localized VGlut2/PSD95 synaptic puncta in the S/Z also revealed a significant increase in thalamocortical synapses in hevin-injected cortices compared to the vehicle-injected littermates (*Figure 4G,H*). By contrast, the number of co-localized VGlut1/PSD95 synaptic puncta was not affected by hevin injection (*Figure 4—figure supplement 1*). Combined with our in vitro data, these in vivo experiments demonstrate that hevin specifically induces thalamocortical synaptic connectivity without affecting intracortical connectivity.

## Loss of hevin leads to morphological immaturity of dendrites and mislocalization of excitatory synapses to dendritic shafts

In the cortex, the majority of excitatory synaptic contacts are compartmentalized onto submicron structures called dendritic spines (*Harris and Kater, 1994*). Cortical dendritic spines, including those in V1, follow a stereotypic maturation timeline (*Irwin et al., 2001*). Long, highly motile filopodia-type protrusions abundant in early development give way to short, stable, wide-headed mushroom spines in the mature brain (*Figure 5—figure supplement 1A*) (*Kaneko et al., 2012*). The spine maturation timeline coincides to a large extent with the expression of hevin protein, which reaches its highest levels at P25 (*Figure 1A,B*). This observation prompted the following question: Does the deficient thalamocortical connectivity in hevin KO coincide with aberrant synaptic morphology? In order to address this question, we investigated whether hevin is involved in the structural development of spine synapses. To do so, we analyzed dendritic morphology in the secondary and tertiary dendrites of layer II/III pyramidal neurons, which receive the majority of the excitatory connections within the S/Z. Analyses of spines in V1 of littermate hevin KO and WT mice at P25 by Golgi-Cox staining showed a significant increase in immature filopodia-like protrusions in P25 hevin KOs concomitant with a decrease in mature mushroom spines compared to littermate WT controls (*Figure 5—figure supplement 1B*). There was no significant difference in total protrusion density between genotypes (WT, 1.12 ± 0.03 spines/µm; KO, 1.07 ± 0.03 spines/µm; *n* = 45 dendrites per condition; p > 0.05, Student's *t* test). The dendritic arborization of layer II/III neurons was also similar between hevin KO and WT mice, showing that lack of hevin does not lead to overt problems in dendritic morphology (*Figure 5—figure supplement 1C*). These results indicated that the astrocyte-secreted synaptogenic protein hevin is important for spine maturation in the cortex.

To understand the role of hevin in dendritic spine maturation at ultra-high resolution, we next employed ssEM in littermate P25 WT and hevin KO mice (*Kuwajima et al., 2013*). Three-dimensional (3D)-EM reconstructions, visualizing dendrites, spines and synapses, confirmed the structural immaturity of hevin KO dendrites (*Figure 5A*). Analysis of postsynaptic density (PSD) area revealed smaller, asymmetric (i.e., excitatory) synapse area in hevin KO V1 (*Figure 5B*), indicative of synaptic immaturity. Despite the deficits in synapse morphology, overall asymmetric synapse density was not significantly affected in hevin KO V1 (WT, 2.16 ± 0.16 synapses/µm; KO, 2.63 ± 0.27 synapses/µm; *n* = 12 dendrites per condition; p = 0.15, Student's *t* test). Since hevin is primarily expressed and secreted by astrocytes, we postulated that hevin KOs may have altered astroglial contact at synapses, but no difference in astrocyte contact was found between WT and hevin KO synapses (*Figure 5C*). Interestingly, a fraction of excitatory synapses in the hevin KOs was made directly onto the dendritic shafts rather than on spines (*Figure 5D*), a configuration that was rare in the WTs at P25. This observation reveals that hevin is required for the proper compartmentalization of excitatory synapses onto spines. Taken together,

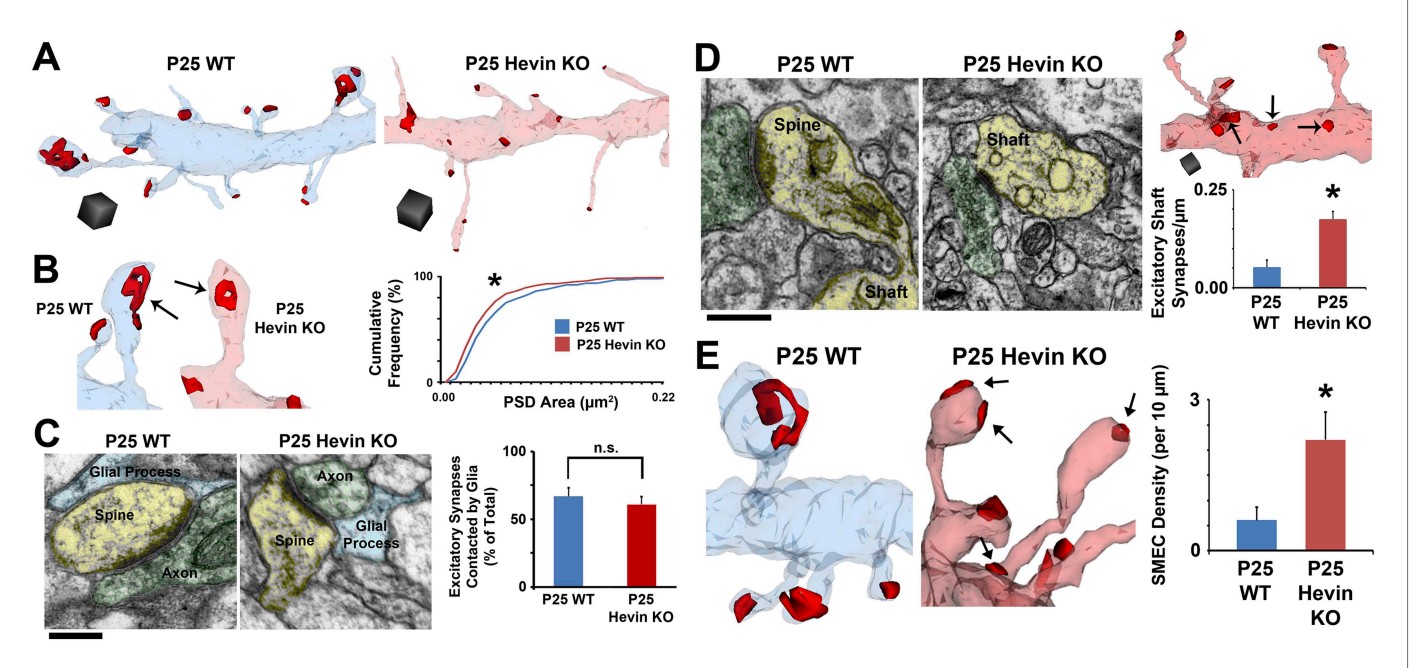

**Figure 5**. Hevin is required for dendritic maturation and proper localization of excitatory synapses. (**A**) Example 3D reconstructions of dendrites in P25 WT (left, blue) and hevin KO (right, pink) V1. Asymmetric PSD locations (i.e., excitatory synapses) are shown in red. Scale cubes, 0.5 μm³. (**B**) Asymmetric PSD size was decreased in P25 hevin KO vs WT (n = 278 WT synapses, 293 KO synapses; p < 0.025, Kolmogorov–Smirnov two-sample test). (**C**) Left: EM examples from P25 WT and P25 hevin KO show the 'tripartite synapse' arrangement of postsynaptic dendritic spines (yellow), presynaptic axonal boutons (green) and glial processes (blue). Scale bar, 250 nm. Right: Quantification revealed no difference in the percentage of excitatory synapses contacted by glial processes in P25 WT vs P25 hevin KO (n = 4 dendrites per animal, 2 animals per genotype; p = 0.52, Student's t test). (**D**) Left: Excitatory synapses, made by axons (green) onto dendritic spines (yellow) in P25 WT, were readily seen on dendritic shafts (yellow) in the hevin KO. Scale bar, 250 nm. Right, Top: Example hevin KO dendrite with multiple excitatory shaft synapses (arrows). Scale cube, 0.5 μm³. Right, Bottom: Quantification of excitatory shaft synapse density in P25 hevin KO compared to P25 WT (n = 4 dendrites per animal, 3 animals per genotype; p < 0.01, Student's t test). (**E**) SMEC density was increased in P25 hevin KO compared to WT (arrows indicate excitatory PSDs on SMECs; n = 4 dendrites per animal, 3 animals per genotype; p < 0.01, one-way ANOVA with Fisher's LSD posthoc test).

The following figure supplement is available for figure 5:

**Figure supplement 1**. Structural immaturity across multiple spine types in hevin KO V1.

these findings show that hevin function is required for the proper maturation and localization of excitatory synapses in the cortex.

## Spines with Multiple Excitatory Contacts (SMECs) represent a stage in excitatory synaptic maturation

Our 3D analyses revealed that, in addition to the above-mentioned structural deficits in the excitatory synapses, a considerable number of dendritic spines receive more than one excitatory synapse in hevin KOs (*Figure 5E*). These SMECs (*Spines with Multiple Excitatory Contacts*) are distinctly different from branched spines, in which multiple spine heads are connected to the same spine neck (*Kirov et al., 1999*). Furthermore, SMECs should not be confused with multisynaptic boutons (MSBs), where a single presynaptic axonal bouton makes contact with multiple dendritic spines (*Kirov et al., 1999*). SMEC density was significantly higher in P25 hevin KO mice compared to WT (*Figure 5E*).

Because hevin KO dendrites displayed other signs of immaturity, we postulated that SMECs may represent an earlier stage in excitatory synapse maturation. To investigate if SMECs occur in the context of normal synaptic development, we created ssEM-derived 3D reconstructions of dendrites in the synaptic zone of WT V1 at P14, an age when dendritic spine structures are not yet fully mature. Electron micrographs revealed the existence of SMECs in P14 V1 in which a single postsynaptic spine contained more than one asymmetric PSD (*Figure 6A*). 3D reconstructions from ssEM confirmed that

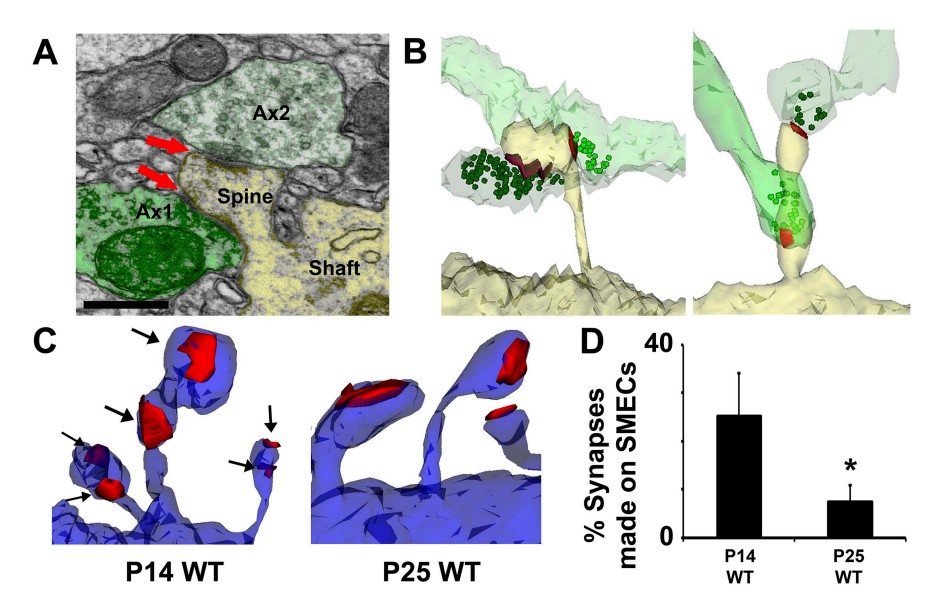

**Figure 6**. Spines with Multiple Excitatory Contacts (SMECs) represent a developmental stage in the maturation of dendritic spine structures. (**A**) Electron micrograph of a SMEC: a dendritic mushroom spine (yellow) making asymmetric contacts (red arrows) with two different axonal boutons (green). Scale bar, 0.5 µm. (**B**) Different spatial arrangements observed in SMECs. Small green circles denote the location of glutamatergic vesicles within the axons. (**C**) SMECs (arrows) decrease from P14–P25 in WT. (**D**) Quantification of the percentage of excitatory synapses made onto SMECs in P14 WT and P25 WT mice (n = 4 dendrites per animal, 3 animals per age; p < 0.05, one-way ANOVA with Fisher's LSD posthoc test).

each PSD on a SMEC was contacted by a different presynaptic axon (*Figure 6B*). This ruled out SMECs as having either a single perforated PSD or multiple PSDs opposed to the same axon. Several configurations of SMECs were detected; some in which two axons synapsed on opposite sides of the same spine head (*Figure 6B*, left), and others with one PSD on the head and a second PSD on either the neck or base of the spine (*Figure 6B*, right). SMECs were primarily of the thin spine type, though we also found numerous examples of filopodia and mushroom SMECs. Remarkably, 25% of all excitatory connections are formed onto SMECs at P14, a finding that may have gone unnoticed if not for the spatial resolution offered by 3D ssEM. The prevalence of SMECs is largely decreased by P25 (*Figure 6C,D*), indicating that SMECs represent a transient stage in excitatory synaptic maturation.

## SMECs are targeted by specific axonal populations

SMECs are a transient structure observed during normal synaptic development, but what purpose do they serve? When multiple axons are contacting a single spine, they will have to share the postsynaptic machinery available within that spine. Such a configuration would potentially provide means to drive competition between neighboring inputs for postsynaptic resources, and the activity levels of the presynaptic axons contacting that spine could directly influence this competition. To determine whether specific axonal populations were contacting SMECs, we completed full 3D reconstructions of axons contacting SMECs in P25 hevin KOs and P14 and P25 WTs. We found that if an axon made a connection with a SMEC, it had nearly a 50% chance of contacting another SMEC nearby (42.5% in P14 WT, 45.4% in P25 WT, 41.6% in P25 hevin KO; *Figure 7A*, *Video 1*), which is vastly higher than what would be expected by chance (6.4% in P14 WT, 0.5% in P25 WT, 3.1% in P25 hevin KO). This specific preference of certain axons for SMECs suggested that SMECs are targeted by particular subpopulations of axons which may be in competition with their neighbors for common postsynaptic spines.

We found that SMECs are still abundant in P25 hevin KOs (*Figure 5E*), and in hevin KOs the number of VGlut2 synapses are reduced whereas VGlut1 synapses are increased at this age (*Figure 2B,C*). Therefore, we postulated that SMECs may be sites for simultaneous innervation by VGlut1+ cortical axons and the VGlut2+ thalamic projections (*Figure 2A*). To provide evidence for this possibility, we

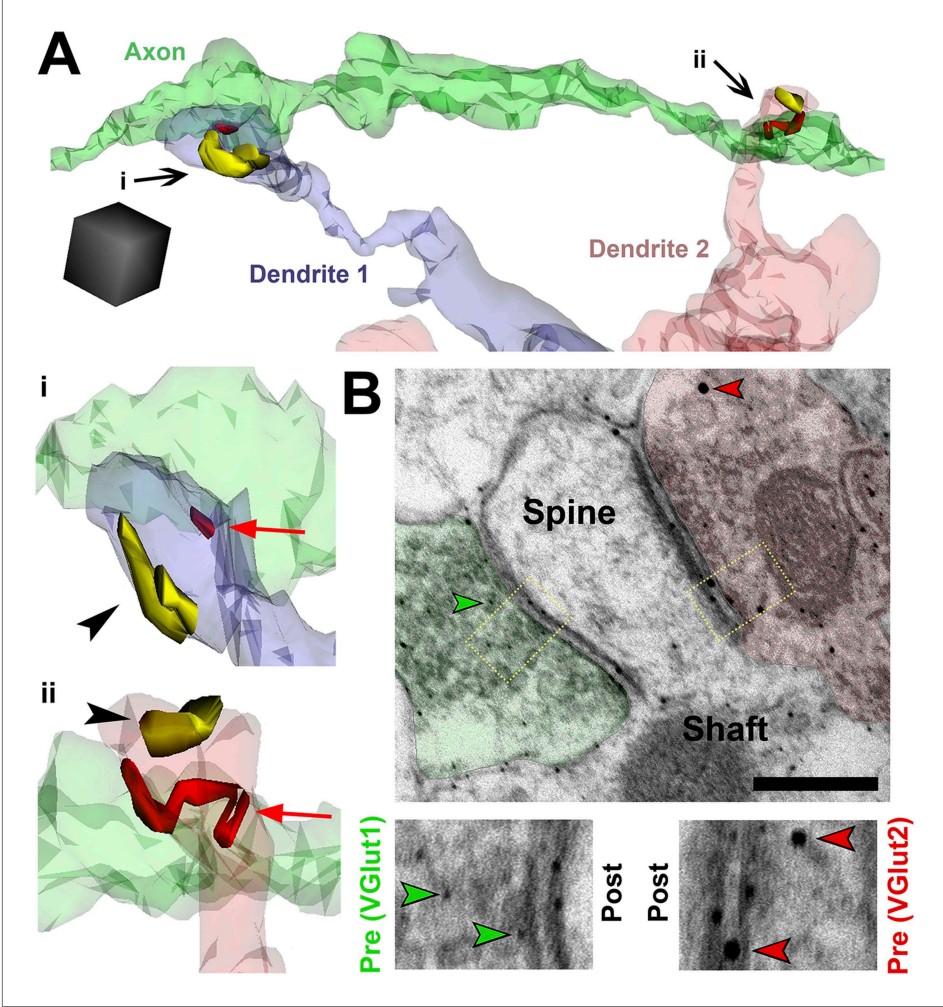

**Figure 7**. SMECs represent potential sites for competition between thalamocortical and intracortical projections. (**A**) Reconstructed axon (green) from P25 hevin KO contacting multiple SMECs on two different dendrites. Quantification revealed that axons that synapsed with at least one SMEC also synapsed with another SMEC roughly 50% of the time. Scale cube, 0.5 μm³. **i–ii**, Zoomed-in images reveal that each SMEC makes an excitatory synapse (red; arrow) with the reconstructed axon as well as a second excitatory synapse (yellow; arrowhead) with an additional axon (not shown). (**B**) Immuno-EM image from V1 in P14 WT showing a SMEC simultaneously contacting a VGlut1+ (green) and VGlut2+ (red) axonal bouton. Higher magnification images (below) highlight the size difference between the small VGlut1 (green arrowheads) and large VGlut2 (red arrowheads) Nanogold particles. Scale bar, 250 nm.

next performed immuno-EM analysis using VGlut1 and VGlut2-specific antibodies on conventional 2D-EM sections from P14 WT mice and observed that a SMEC can indeed receive simultaneous cortical (VGlut1+) and thalamic (VGlut2+) inputs (*Figure 7B*).

It was previously shown that, during early cortical development, VGlut2+ thalamic inputs establish numerous contacts within the S/Z; however, as the cortex develops, the majority of these connections are pruned and VGlut1+ inputs dominate this region (*Miyazaki et al., 2003*). During this period, confocal microscopy images of VGlut1+ and VGlut2+ terminals first appear to frequently overlap, but this overlap resolves over time (*Nakamura et al., 2007*). Based on our findings, we postulated that the close positioning of VGlut1+ and VGlut2+ terminals at the same SMEC might yield an apparent co-localization between these two presynaptic markers in conventional light microscopy. If that is the case we expected hevin to affect the segregation of VGlut1 and VGlut2 terminals during cortical development. To test this possibility, we used IHC to compare the apparent co-localization of the two VGluts in the S/Z of V1 in WTs and hevin KOs at P15 and P25. We found that, in agreement with earlier

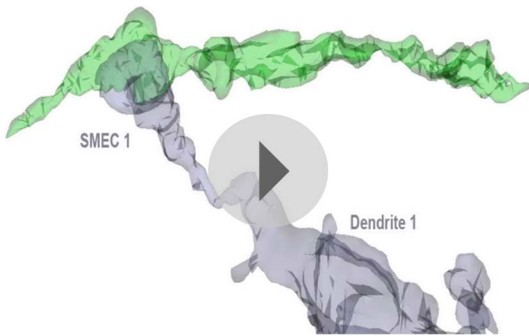

**Video 1**. An axon (green) from Layer II/III V1 reconstructed in 3D from ssEM images of P25 hevin KO. SMEC 1 from Dendrite 1 (blue) makes Synapse 1 (red) with this axon and Synapse 2 (yellow) with a second unshown axon. On a different part of this same axon, it makes Synapse 3 (red) with SMEC 2 from Dendrite 2 (pink) that also contacts a third (also unshown) axon at Synapse 4 (yellow).

findings from other brain regions and layers (*Herzog et al., 2006*; *Nakamura et al., 2007*), 10.97% of VGlut terminals appeared to co-localize in P15 WT, which was significantly reduced in P25 WT (*Figure 8—figure supplement 1A,B*). In hevin KOs, co-localization of VGlut1/VGlut2 was originally lower than WT at P15, perhaps due to the overall loss of VGlut2+ synapses. Interestingly, the frequency of overlap in the P25 hevin KO was essentially unchanged compared to P15 and was significantly higher than the P25 WT (*Figure 8—figure supplement 1A,B*). These results suggest that hevin is important for the resolution of the VGlut1/VGlut2 overlap (and, potentially, SMECs). Analysis of rotated images confirmed that the observed VGlut1/VGlut2 overlap by confocal microscopy is not due to random chance (*Figure 8—figure supplement 1C*).

The VGlut1/VGlut2 overlap has previously been attributed to temporary co-expression of these proteins in the same synaptic terminals (*Nakamura et al., 2007*). To test if this co-localization is mainly due to the resolution limit of light microscopy rather than the expression of these two VGluts at the same terminal, we used high-resolution structured illumination microscopy (SIM) imaging of presynaptic puncta in the synaptic zones of P15 WT mice (since the VGlut1/2 overlap is highest at this age). SIM uses constructive and deconstructive interference of excitation light at the focal plane of the objective to illuminate a sample with a series of sinusoidal stripes; from the resulting moiré fringes it is possible to generate super-resolution data and produce a minimum two-fold improvement in resolution over confocal microscopy (*Schermelleh et al., 2010*). Indeed, SIM imaging of VGlut1 and VGlut2 puncta in P15 WT showed virtually no co-localized presynaptic puncta (*Figure 8—figure supplement 1D*). By increasing the maximum distance with which to detect co-localized puncta, SIM analysis eventually reached a point at which VGlut1/VGlut2 overlap approached the level observed in P15 WT by confocal imaging; this occurred near the practical resolution of confocal microscopy (*Figure 8—figure supplement 1E*). This provided strong evidence that the overlap of VGlut1/VGlut2 puncta we observed with confocal microscopy (*Figure 8—figure supplement 1A,B*) was due to adjacent presynaptic puncta rather than co-expression in the same terminal.

Despite the knowledge that SMECs are increased in the hevin KO (*Figure 5E*) and can receive simultaneous VGlut1/VGlut2 synaptic inputs (*Figure 7B*), the vast majority of inputs to hevin KO cortex remain VGlut1-positive (*Figure 2B*). Therefore, the possibility arises that most SMECs actually receive inputs from multiple intracortical axons, rather than existing as sites for simultaneous thalamic and cortical innervation. Taking advantage of the finding that co-localized VGlut1/VGlut2 puncta in light microscopy is predominantly due to close expression by different presynaptic terminals (*Figure 8—figure supplement 1D,E*), we next imaged presynaptic puncta in close proximity to dendritic spines with confocal microscopy in order to quantify the types of inputs made onto SMECs. Using an in utero electroporation (IUE) approach at embryonic day 15.5 (E15.5) we specifically labeled cortical layer II/III neurons with green fluorescent protein (GFP) (*Figure 8—figure supplement 2A,B*). We then harvested brains at specific developmental ages and co-stained for VGlut1 and VGlut2. S/Z dendrites and presynaptic puncta were imaged via confocal Z-stacks and reconstructed in 3D with the Imaris processing package (*Figure 8—figure supplement 2C,D*). Presynaptic puncta in close proximity to the dendrite were identified using a Matlab algorithm embedded in Imaris (see 'Materials and methods'). This analysis method was first confirmed in P15 WT, an age when SMECs are still prevalent. Unisynaptic spines (i.e., spines contacting only one presynaptic puncta; either VGlut1 or VGlut2) and SMECs in various configurations were successfully detected (*Figure 8—figure supplement 2D,E*). In accordance with a role for SMECs in synaptic competition during development, the majority of SMECs were innervated by both VGlut1 and VGlut2 inputs at this age, with fewer examples of VGlut1/VGlut1 and VGlut2/VGlut2 SMECs (*Figure 8—figure supplement 2E*).

Using the same parameters, we then quantified the difference in presynaptic innervation of SMECs between WT and hevin KO V1 at P21. In addition to confirming the overall increase in SMECs in the hevin KO, the 3D reconstructions revealed that the majority of these SMECs indeed received mixed VGlut1/VGlut2 inputs (*Figure 8A,B*), rather than having multiple VGlut1 inputs as the overall presynaptic puncta density would suggest. Taken together, these findings reveal that SMECs are typically sites of contact by cortical and thalamic axons, potentially representing a novel paradigm for synaptic competition between these two main sources of presynaptic input onto cortical spines. Our results

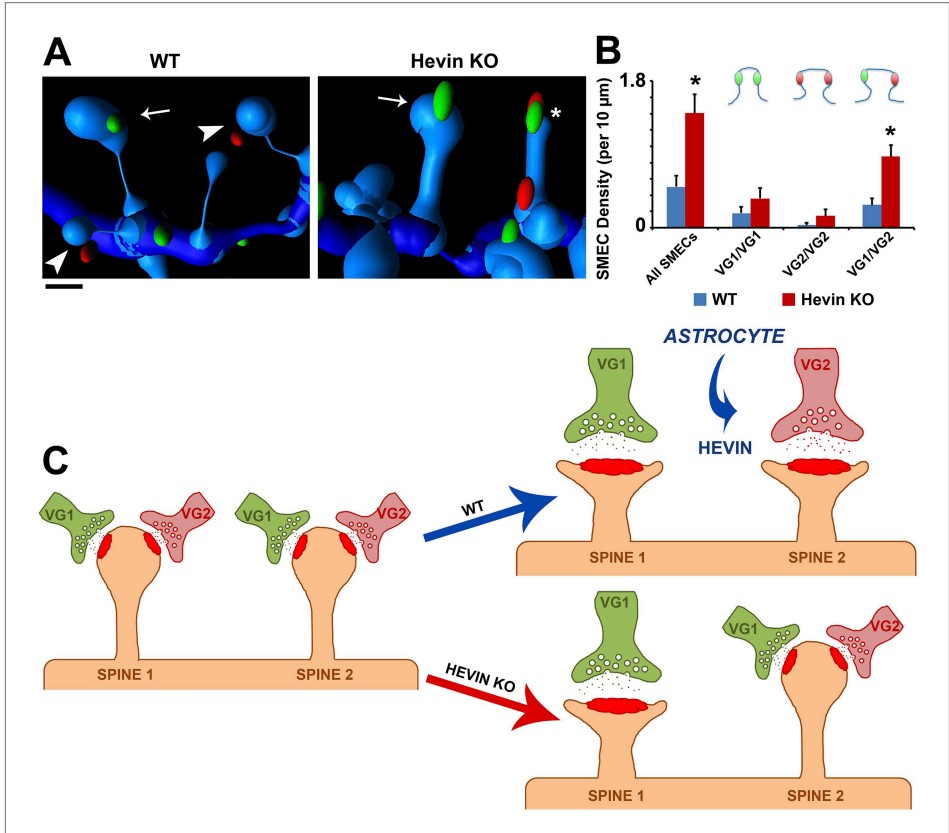

**Figure 8**. Hevin is critical for the resolution of VGlut1/VGlut2-innervated SMECs. (**A**) Representative Imaris 3D reconstructions of GFP-labeled dendrites (blue) in the S/Z of P21 WT and hevin KO V1. Presynaptic puncta are rendered as elliptical 'spots' (green = VGlut1; red = VGlut2). Numerous unisynaptic spines can be seen in the WT (arrow = VGlut1 spine; arrowheads = VGlut2 spines). In hevin KO, a SMEC can be seen contacting both a VGlut1 and VGlut2 'spot' (asterisk). A unisynaptic VGlut1 spine (arrow) is also present. Scale bar, 0.5 µm. (**B**) Quantification of SMEC density at P21 shows that the increase in total SMECs in the hevin KO is driven by the VGlut1/VGlut2 SMEC subtype (3 animals/genotype, n = 15 dendrites per condition; p < 0.01, Student's *t* test). (**C**) Model for astrocytic control of thalamocortical connectivity by hevin. Left: In early V1 synaptic development, intracortical (primarily VGlut1, or VG1) axons compete with thalamocortical (primarily VGlut2, or VG2) axons for synapses, occasionally forming synapses on the same dendritic spine (resulting in a SMEC). In the WT, astrocytes secrete hevin which stabilizes VGlut2+ synapses, resulting in discrete populations of VGlut1+ and VGlut2+ unisynaptic spines. In the hevin KO, VGlut2+ synapses cannot be properly stabilized. These sites either remain in competition with VGlut1, explaining the persistence of SMECs in hevin KO, or become lost, resulting in more VGlut1+ synapses overall.

The following figure supplements are available for figure 8:

**Figure supplement 1**. Overlap of VGlut1 and VGlut2 in light microscopy is due to close proximity of different presynaptic terminals.

**Figure supplement 2**. Imaging presynaptic terminals in proximity to dendritic spines.

indicate that resolution of SMECs into single synapse spines is a developmental process that is regulated by astrocytes through secretion of hevin.

## Discussion

Synaptic maturation and refinement, wherein appropriate synapses are strengthened while superfluous connections are eliminated, are critical for the establishment of functional neuronal circuitry. In the developing cortex, the process of refinement includes the resolution of synaptic competition between incoming thalamic projections and the resident intracortical axons. In the present study, we show that an astrocyte-secreted synaptogenic protein, hevin, is required for the connectivity of thalamocortical synapses. Furthermore, we found that thalamic and cortical axons simultaneously innervate dendritic spines, uncovering a potential role for spines as sites of synaptic competition. Based on our findings, we propose that, in the developing visual cortex, SMECs serve as sites at which cortical and thalamic inputs compete (*Figure 8C*). This competition is normally resolved by P25 with the establishment of single synapse spines contacting either VGlut1+ or VGlut2+ terminals. This resolution is dependent on astrocyte-secreted hevin to stabilize VGlut2+ synapses. In the absence of hevin, the VGlut2+ connections cannot be stabilized thus cannot effectively compete with VGlut1+ terminals for postsynaptic spines, resulting in increased VGlut1+ synapses and the persistence of SMECs (*Figure 8C*).

Since hevin is a synaptogenic protein, our initial findings showing no overall reduction in excitatory synaptic density in area V1 of hevin KO mice were surprising. However, this observation was explained by a severe and specific reduction of in the number of thalamocortical connections in hevin KOs, whereas the abundance of intracortical synapses was increased. These results reveal that during development the formation and stabilization of VGlut1+ intracortical and VGlut2+ thalamocortical synapses are differentially regulated. Our findings also suggest that the reduction in VGlut2+ synapses in hevin KO V1 enable synaptic takeover by the more abundant VGlut1+ projections. In agreement with this possibility, previous observations of synaptic competition have shown that retraction of one axon is often succeeded by the expansion of another (*Walsh and Lichtman, 2003*).

The specificity of hevin for VGlut2+ synaptic stabilization raises the question of why we observed global structural immaturity of dendrites in hevin KO V1. During development, spontaneous retinal activity relayed by thalamus facilitates proper patterning and connectivity within visual cortex (*Ackman and Crair, 2014*). Furthermore, it was previously shown that perturbations in synaptic competition for eye-specific territories in V1 can compromise dendritic spine refinement (*Mataga et al., 2004*). Defective thalamocortical connectivity in hevin KO mice may therefore be responsible for driving generalized spine synapse immaturity in the cortex. In agreement with this possibility, we found that a hevin KOs had increased numbers of excitatory synapses made onto dendritic shafts. It is postulated that excitatory shaft synapses are among the first synapses to form along dendrites (*Reilly et al., 2011*). These shaft synapses may be induced to become spines by presynaptic activity during the early stages of postnatal development (P8–P12) (*Kwon and Sabatini, 2011*); our results suggest that thalamocortical connectivity in general and/or hevin in particular may be playing a role in the shaft to spine transition of some synapses.

Hevin has been known to modulate cell adhesion (*Sullivan and Sage, 2004*), and its positioning in the synaptic cleft (*Lively et al., 2007*) makes it a prime candidate for organizing and stabilizing pre- and postsynaptic cell adhesion molecules. Because hevin is preferentially inducing thalamocortical synapses, there may be specific hevin interactors present on these synapses that distinguish them from the intracortical ones. Hevin may also act as a 'protection signal' (*Sanes and Lichtman, 1999*) to prevent elimination of thalamocortical synapses. Alternatively, hevin is a generalized protective signal but VGlut1+ intracortical synapses may have redundant stabilization/protection mechanisms in place that can compensate for the lack of hevin, whereas with no such compensation available the VGlut2+ thalamocortical population is lost.

Our investigation of hevin KO cortex presents an interesting possibility that cortical and thalamic axons compete for postsynaptic targets at single dendritic spines. The presence of SMECs in the central nervous system was described a long time ago (*Jones and Powell, 1969)*. Perhaps due to the difficulty of observing these synaptic structures, because of their unique geometrical arrangement, this early finding of SMECs was largely ignored and their purpose remained unknown. Instead, a simple 'one spine: one excitatory synapse' view of connectivity prevailed. The one spine: one synapse configuration provides a context in which spines can compartmentalize calcium and filter membrane

potentials in an input-specific manner (*Yuste, 2013*). Though important, these features of spines may not represent the full extent of their functions in the developing CNS.

Our findings indicate that SMECs represent a stage in excitatory synaptic development. In agreement with this, multiply-innervated filopodia have been seen in the juvenile CA1 region of hippocampus (*Fiala et al., 1998*); these protrusions were proposed to 'sample' alternative axonal partners during a highly active period of synaptogenesis. In addition, live imaging of green fluorescent protein (GFP)-labeled dendrites in acute hippocampal slices from P10–12 mice showed that spines sample nearby synaptic resources even after making a stable contact (*Konur and Yuste, 2004*). This sampling was proposed to trigger synaptic competition at individual spines. However, these experiments inferred the presence of axonal boutons from labeling of the presynaptic marker synaptophysin, raising the possibilities that this staining could have been the result of multiple presynaptic release sites (i.e., active zones) on the same synapse or even different boutons belonging to the same axon. By using ssEM-derived 3D reconstructions of synaptic structures, here we show that multiple independent axons compete for synaptic territory on single spines. Furthermore, our finding that SMECs are contacted by different axonal populations (i.e., VGlut1+ and VGlut2+) suggests that establishment of synaptic networks in the cortex depends on the outcome of synaptic competition at spines, demonstrating a dynamic new role for spines in synaptic development.

In conclusion, here we show that the astrocyte-secreted hevin is required for the proper establishment of thalamocortical synapses. Moreover, this process occurs on spines serving as simultaneous contact sites for thalamic and cortical inputs during development, a finding that expands the current view of spines as input filters or calcium buffers. These results may also have important clinical implications. Hevin is strongly upregulated in reactive astrocytes in disease conditions (*McKinnon and Margolskee, 1996*) and has also been linked to neurological disorders, including autism, schizophrenia, suicide and depression (*Purcell et al., 2001*; *Jacquemont et al., 2006*; *Kahler et al., 2008*; *Vialou et al., 2010*; *Zhurov et al., 2012*; *De Rubeis et al., 2014*). Abnormal spine maturation and connectivity have also been observed in these and other diseases (*Fiala et al., 2002*; *De Rubeis et al., 2013*; *Kim et al., 2013*), including the presence of SMEC-like 'giant spines' in hippocampi from patients with severe temporal lobe epilepsy (*Witcher et al., 2010*). Future studies may determine if impaired resolution of axonal competition by astrocytes drives the dendritic spine deficiencies observed in these conditions, providing a novel cellular target for therapeutic strategies.

## Materials and methods

### Immunohistochemistry and synaptic puncta analysis

For synaptic puncta analysis of mouse V1, hevin KO mice on a 129/Sve background and littermate age-matched WT controls were perfused intracardially with Tris-Buffered Saline (TBS, 25 mM Tris-base, 135 mM NaCl, 3 mM KCl, pH 7.6) supplemented with 7.5 µM heparin followed with 4% paraformaldehyde (PFA; Electron Microscopy Sciences, PA) in TBS. The brains were then removed and were fixed with 4% PFA in TBS at 4°C overnight. The brains were cryoprotected with 30% sucrose in TBS overnight and were then embedded in a 2:1 mixture of 30% sucrose in TBS:OCT (Tissue-Tek, Sakura, Japan). Brains were cryosectioned (coronal) at 20 µm using Leica CM3050S (Leica, Germany). Sections were washed and permeabilized in TBS with 0.2% Triton-X 100 (TBST; Roche, Switzerland) three times at room temperature. Sections were blocked in 5% Normal Goat Serum (NGS) in TBST for 1 hr at room temperature. Primary antibodies (guinea pig anti-VGlut1 1:3500 [AB5905, Millipore, MA], guinea pig anti-VGlut2 1:7500 [135 404, Synaptic Systems, Germany], rabbit anti-VGLUT2 1:750 [135 403, Synaptic Systems], rabbit anti-PSD95 1:300 [51–6900, Invitrogen, CA]) were diluted in 5% NGS containing TBST. Sections were incubated overnight at 4°C with primary antibodies. Secondary Alexa-fluorophore conjugated antibodies (Invitrogen) were added (1:200 in TBST with 5% NGS) for 2 hr at room temperature. Slides were mounted in Vectashield with DAPI (Vector Laboratories, CA) and images were acquired on a Leica SP5 confocal laser-scanning microscope.

3–5 animals/genotype/age were stained with pre- (VGlut1 or VGlut2) and post-synaptic (PSD95) marker pairs as described previously (*Kucukdereli et al., 2011*). Three independent coronal sections per each mouse, which contain the V1 visual cortex (Bregma −2.5−−3.2 mm, Interaural 1.3–0.6 mm [*Franklin and Paxinos, 2001*]) were used for analyses. 5 µm thick confocal z-stacks (optical section depth 0.33 µm, 15 sections/z-stack, imaged area/scan = 20,945 µm²) of the synaptic zone in area V1 were imaged at 63× magnification on a Leica SP5 confocal laser-scanning microscope. Maximum

projections of three consecutive optical sections (corresponding to 1 µm total depth) were generated from the original z-stack. Analyses were performed blind as to genotype. The Puncta Analyzer plugin that was developed by Barry Wark (available *Source code 1*) for ImageJ 1.29 (NIH; http://imagej.nih.gov/ij/, version ImageJ 1.29 is available at http://labs.cellbio.duke.edu/Eroglu/Eroglu_Lab/Publications.html) was used to count the number of co-localized, pre-, and post-synaptic puncta. This quantification method is based on the fact that pre- and post-synaptic proteins (such as VGluts and PSD95) are not within the same cellular compartments and would appear co-localized only at synapses due to their close proximity. Previous studies showed that this quantification method yields an accurate estimation of the number of synapses in vitro and in vivo which were previously confirmed by other methods such as EM and electrophysiology by us and others (*Christopherson et al., 2005*; *Eroglu et al., 2009*; *Kucukdereli et al., 2011*). Details of the quantification method have been described previously (*Ippolito and Eroglu, 2010*). Briefly, 1 µm thick maximum projections are separated into red and green channels, background subtracted (rolling ball radius = 50), and thresholded in order to detect discrete puncta without introducing noise. The Puncta Analyzer plugin then uses an algorithm to detect the number of puncta that are in close alignment across the two channels, yielding quantified co-localized puncta. In order to calculate % of WT co-localization, co-localized puncta values for WT were averaged, then all image values (WT and KO) were converted to % of the calculated WT average.

For co-localization of VGlut1 and VGlut2, three P25 hevin KO mice on a 129/Sve background, three littermate P25 WT controls and three P15 WT controls were perfused, sectioned, stained and imaged as described previously for synaptic staining. Primary antibodies (guinea pig anti-VGLUT1 1:3500 [AB5905, Millipore], rabbit anti-VGLUT2 1:750 [135 403, Synaptic Systems]) were diluted in 5% NGS containing TBST.

For cell staining, three P25 WT mice and one Aldh1L1-eGFP mouse (in which astrocytes are labeled with eGFP; MMRRC, UC Davis, CA) were perfused and sectioned as described previously for synaptic staining. Sections containing Layer II/III of V1 visual cortex (Bregma −2.5–−3.2 mm, Interaural 1.3 to 0.6 mm [*Franklin and Paxinos, 2001*]) or dorsal LGN (Bregma −1.7–−2.9 mm, Interaural 2.1–0.9 mm [*Franklin and Paxinos, 2001*]) were washed and permeabilized in TBS with 0.2% Triton-X 100 (TBST; Roche) three times at room temperature. Sections were blocked in 5% Normal Donkey Serum (NDS) in TBST for 1 hr at room temperature. Primary antibodies (mouse anti-GFAP 1:1000 [G3893, Sigma, MO], mouse anti-NeuN clone A60 1:1000 [MAB377, Millipore], rabbit anti-Iba1 1:500 [019–19,741, Wako, Japan], goat anti-hevin [a.k.a. SPARCL1] 1:500 [AF2836, R&D Systems, MN]) were diluted in 5% NDS containing TBST. Sections were incubated overnight at 4°C with primary antibodies. Secondary Alexa-fluorophore conjugated antibodies (Invitrogen) were added (1:200 in TBST with 5% NDS) for 2 hr at room temperature. Slides were mounted in Vectashield with DAPI (Vector Laboratories) and images were acquired at 63× magnification on a Leica SP5 confocal laser-scanning microscope.

## Western blotting

WT mice were perfused with PBS intracardially to clear blood before the brains were removed. Cortex and hippocampus were dissected out and homogenized in ice-cold solubilization buffer (25 mM Tris pH 7.2, 150 mM NaCl, 1 mM $CaCl_2$, 1 mM $MgCl_2$) containing 0.5% NP-40 (Thermo Scientific, MA) and protease inhibitors (Roche). The protein concentrations of the lysates were determined using micro BCA protein assay kit (Pierce, IL). Samples for SDS-PAGE were prepared at 1 µg protein/µl concentration using 5× SDS-PAGE buffer (Pierce). 10 µg of protein was loaded into each well. Samples were resolved by SDS-PAGE on 4–15% polyacrylamide gels (BioRad, CA) and were transferred onto an Immobilon-FL PVDF membrane (Millipore).

Blots were blocked in 50% fluorescent blocking buffer in PBS (MB-070, Rockland, PA) containing 0.01% Tween-20 for 1 hr at room temperature. Blots were then incubated with primary antibody dilutions in blocking buffer (goat anti-SPARCL1 1:2000 [AF2836, R&D Systems], rabbit anti-β-tubulin 1:1000 [926–42,211, Li-Cor, NE]) overnight at 4°C. Fluorescently labeled secondary antibodies (Li-Cor) were diluted (1:5000) in the same buffer as primary antibodies and western blots were incubated with secondary antibodies for 2 hr at room temperature in the dark. Detection was performed using the Li-Cor Odyssey System.

## Electrophysiology

Acute coronal slices (350 µm thick) were prepared from P23–P26 Hevin-null animals and littermate wildtype controls. Mice were deeply anesthetized with tribromoethanol (Alfa Aesar, MA) and

transcardially perfused with ice-cold high-sucrose artificial cerebrospinal fluid (ACSF) equilibrated with 95% $O_2$ and 5% $CO_2$ (carboxygenated). Brains were removed and sectioned in ice-cold sucrose ASCF on a Leica VT1200S. Slices recovered in carboxygenated standard ACSF at room temperature for a minimum of 1 hr. Whole-cell patch-clamp recordings of miniature excitatory postsynaptic currents (mEPSCs) were recorded at 30°C in ACSF supplemented with tetrodotoxin (Tocris, UK) and picrotoxin (Sigma), with a continuous perfusion rate of 2–3 ml/min. Membrane potential was held at −70 mV. Junction potential was uncorrected. Recording pipette internal solution (pH 7.2) contained (in mM): 103 CsOH, 103 D-gluconic acid, 2.8 NaCl, 5 TEA-Cl, 20 HEPES, 0.2 EGTA, 5 Lidocaine N-ethyl chloride, 4 ATP-Mg, 0.3 GTP-Na, 10 $Na_2$ Phosphocreatine, 0.025 Alexa 488 Hydrazide (Invitrogen) and approximately 10 $K_2SO_4$ (to bring solution to 300 mOsm).

Cells from Layer II/III of the visual cortex (30–80 microns ventral from the ventral border of Layer I) were visualized with a 40× water-immersion objective (LUMPlanFI, 40×/0.80 water immersion) under an Olympus BX51WI microscope equipped with infrared differential interference contrast optics, reflected fluorescence system, and OLY-150 camera (Olympus, Japan). Signals were recorded using a MultiClamp 700B amplifier and DigiData 1322A (Molecular Devices, CA). Cells with capacitance of less than 100 picofarads were excluded to prevent inclusion of interneurons and pyramidal morphology was confirmed post-recording by visualizing the Alexa 488 dye. Pipette resistance ranged from 2.8–4.1 MΩ. Signals were collected at 10 kHz, unfiltered. Cells with poor technical quality of recordings were excluded from analysis based on predefined criteria of >25 MΩ access resistance or >30% change in series resistance or >6 pA peak-to-peak noise. Miniature EPSC events were analyzed off-line using MiniAnalysis software (Synaptosoft, GA).

## Virus-assisted axonal projection tracing

P18 hevin KO and WT littermate mice were deeply anesthetized with intraperitoneal injection of ketamine (150 mg/kg)/xylazine (15 mg/kg). Using Nanoject (Drummond, PA), mice were stereotactically injected by 50 nl of EF-1a promoter-driven Flex-AAV-GFP within the dLGN [AP: −2.0; ML: 2.0; DV: 2.3 from brain surface]. To visualize specific neurons in dLGN that are directly connected with visual cortex, 100 nl of rabies virus glycoprotein-coated Lenti-FuGB2-Cre (synapsin promoter) (*Kato et al., 2011*) was infected into the V1 region of visual cortex [AP: −3.5; ML: 2.5; DV: 0.3 from brain surface]. 2 weeks after infection, brains were removed, postfixed overnight at 4°C, and then cryo-protected with 30% sucrose in TBS. Brains were cut into 50 μm coronal sections by cryostat (Leica CM 3000). Sections were counterstained with DAPI (Sigma). After washing three times, the sections were coverslipped with FluorSave (CalBioChem, Merck, Germany) aqueous mounting medium. For the axonal fiber tracing, images were taken by tile scan imaging using LSM 710 confocal microscope (Zeiss, Germany) with a 10× objective under control of Zen software (Zeiss).

## Cortical/thalamic cell culture and synapse assay

Neurons from either cortex or thalamus were purified from P1 hevin KO pups by sequential immunopanning as follows: Following dissection, cortex/thalamus was digested for 30 min in papain (Worthington, NJ). Papain digestion was then inhibited in sequential low/high concentrations of ovomucoid inhibitor (Worthington) and the resultant digested tissue was passaged through a 20 μm Nitex mesh filter (Sefar, NY). The cell solutions then underwent negative immunopanning (to remove nontarget cells and debris) on 2× Bandeiraea Simplicifolia Lectin I-coated petri dishes (Vector Laboratories), AffiniPure goat-anti mouse IgG+IgM (H+L) (Jackson Immunoresearch Laboratories, PA) coated dish, and AffiniPure goat-anti rat IgG (H+L) (Jackson) coated dish. A round of positive panning, using rat anti-neural cell adhesion molecule L1 antibody, clone 324 (MAB5272, Millipore), was used to isolate neurons from other cell types (predominantly astrocytes) to greater than 95% purity. Cortical cells or mixed cortical/thalamic cells (1:1 ratio) were then cultured in serum-free medium containing BDNF, CNTF, and forskolin on laminin-coated coverslips as previously described (*Christopherson et al., 2005*; *Kucukdereli et al., 2011*). Recombinant hevin protein was purified as described previously (*Kucukdereli et al., 2011*). Neurons were cultured for 3 days, then were treated for 36 hr with AraC to kill any contaminating mitotic cells (i.e., astroglia), then were cultured with 90 nM hevin or hevin-free growth media for an additional 9 days.

Synapse quantification of cortical/thalamic cultures follows the procedure outlined in *Kucukdereli et al. (2011)* with the exception of the antibodies used: primary antibodies against VGlut1 (1:1000; guinea pig; Millipore), VGlut2 (1:500; rabbit; Synaptic Systems), and PSD95 (1:500; mouse; Neuromab,

CA); secondary antibodies consisted of Alexa-conjugated antibodies diluted 1:1000 in antibody buffer. Imaging was performed on the AxioImager M1 (Zeiss) at 63× magnification. Only cortical neurons were imaged and thalamic neurons were avoided for imaging by the appearance of bright VGlut2 staining within the cell soma.

### In vivo hevin injections

P13 hevin KO and WT littermate mice were deeply anesthetized with intraperitoneal injection of ketamine (150 mg/kg)/xylazine (15 mg/kg). With Nanoject, mice were stereotactically injected with either recombinant hevin protein (200 ng in Dulbecco's PBS, Gibco, CA) or vehicle control (DPBS; 100 nl) into layer II/III of area V1 [AP: −2.1; ML: 2.3; DV: 0.25 from brain surface]. After 3 days, pups were anesthetized with Avertin then perfused transcardially with TBS containing heparin followed by 4% PFA. Brains were harvested, post-fixed overnight in 4% PFA, then cryoprotected in 30% sucrose-TBS. Sections (20 μm) were cut on a cryostat (Leica) and stained for VGlut1/VGlut2/PSD95 as described above. Imaging was performed on a Leica SP5 confocal microscope in area V1 approximately 100 μm laterally to the site of injection.

### Golgi-cox staining, dendritic spine analysis and neuronal morphology

Golgi-cox staining was performed on hevin KO and littermate WT control mice (n = 3 mice per genotype) as described in the FD Rapid GolgiStain Kit (FD NeuroTechnologies, MD). Dye-impregnated brains were embedded in Tissue Freezing Medium (Triangle Biomedical, NC) and were rapidly frozen on ethanol pretreated with dry ice. Brains were cryosectioned coronally at 100 μm thickness and mounted on gelatin-coated microscope slides (LabScientific, NJ). Sections were stained according to the directions provided by the manufacturer.

Three independent coronal sections per each mouse, which contain the V1 visual cortex (Bregma −2.5–−3.2 mm, Interaural 1.3–0.6 mm [*Franklin and Paxinos, 2001*]) were imaged. Layer II/III pyramidal neurons were identified by their distance from pia and their distinct morphologies. Secondary and tertiary dendrites of these neurons were selected for analysis. Z-stacks of Golgi-stained dendrites (up to 80 microns total on z-axis; optical section thickness = 0.5 μm) were taken at 63× magnification on a Zeiss AxioImager M1. Series of TIFF files corresponding to each image stack were loaded into RECONSTRUCT software (*Fiala, 2005*) (freely available at http://synapses.clm.utexas.edu). For each series, 3 × 10 μm segments of dendrites were chosen for analysis. 15 dendrites were analyzed per animal making a total of 45 dendrites per condition. Analyses were performed blind as to genotype. Dendritic spines were identified on the selected dendritic segments; more than 500 spines per genotype were analyzed. Spines were analyzed by the rapid Golgi analysis method described in *Risher et al. (2014)*. Briefly, using the 'Draw Z-trace' tool in RECONSTRUCT, the three-dimensional length of each spine (from the point where the spine neck contacted the dendritic shaft out to the tip of the spine head) was measured. Spine head width was measured by drawing a straight line across the widest point of each spine in a single image of the z-series. These measurements were exported to Microsoft Excel (Microsoft, WA), where a custom macro was used to classify spines based on the spine length, width, and length:width ratio measurements obtained in RECONSTRUCT. Spines were categorized based on the following hierarchical criteria: (1) Branched = more than one spine head attached to same spine neck; (2) Filopodium = length > 2 μm; (3) Mushroom = width > 0.6 μm; (4) Long thin = length > 1 μm; (5) Thin = length:width ratio > 1; (6) Stubby = length:width ratio ≤ 1.

For quantification of neurite outgrowth and branching, 100 μm thick coronal Golgi-cox stained sections were visualized using a Zeiss AxioImager D2 microscope. A total of 24 V1 Layer 2–3 neurons (4 neurons per animal, 3 animals per condition) were selected for dendritic tracing from P25 hevin null and littermate WT controls. Tracing was performed with Neurolucida tracing tool (MBF Bioscience, VT). Convex hull analysis was used to measure total dendritic length and area, while Sholl analysis was used to determine dendritic complexity/branching. All analyses were done with NeuroExplorer (MBF Bioscience) software.

### Serial section electron microscopy

For ssEM analysis of mouse V1, P14 WT controls, P25 hevin KO mice and their littermate P25 WT controls (3 mice per genotype/age, all on a 129/Sve background) were first transcardially perfused with warm PBS solution to clear out blood cells, and then with warm (37°C) 2% PFA, 2.5% glutaraldehyde (EMS), 2 mM $CaCl_2$, and 4 mM $MgCl_2$ in 0.1 M cacodylate buffer (EMS) (pH 7.4) under Tribromoethanol (Sigma) anesthesia. 400 μm thick coronal sections per each mouse, which contain the V1 visual cortex

(Bregma −2.5−−3.2 mm, Interaural 1.3–0.6 mm [*Franklin and Paxinos, 2001*]) were cut with a tissue chopper (Stoelting, IL) and area V1 was dissected out with a #11 scalpel blade. V1 slices were immersed in 2% glutaraldehyde, 2 mM $CaCl_2$, and 4 mM $MgCl_2$ in 0.1 M cacodylate buffer (pH 7.4) and fixed overnight at 4°C. At the Duke Electron Microscopy Service core facility, slices were rinsed 3 × 5 min in 0.1 M phosphate buffer (PB) and postfixed in 1% $OsO_4$ (Sigma) while heating in a microwave (2 min on, 2 min off, then 2 min on at 70% power with vacuum). After rinsing 2 × 5 min with 0.1 M PB, they were dehydrated in ethanol/acetone series enhanced with 40 s of microwave processing. They were next incubated in 50:50 acetone:epoxy overnight at room temperature. After two changes of straight epon 3 × 3 min in the microwave, slices were left to stand for 30 min and then embedded in 100% epon resin at 60°C for 48 hr. Ultrathin serial sections (45–50 nm) were cut from a small trapezoid positioned 50–100 µm below the pial surface corresponding to the synaptic zone (a.k.a. layer I) which contain the dendrites of layer II/III neurons. Serial sectioning, processing and photography were carried out by the Electron Microscopy Core at Georgia Regents University, following a protocol adapted from *Harris et al. (2006)*.

Series consisting of 100–150 consecutive micrographs each were blinded as to condition prior to analysis. Serial sections were aligned and synaptic structures were traced using RECONSTRUCT software. Section thickness was calculated with the cylindrical diameters method (*Fiala and Harris, 2001*). Dendrites were chosen for analysis on the basis of (1) spanning at least 75 consecutive serial sections, (2) measuring between 0.4–0.8 µm in diameter in cross-section (to exclude large, apical dendrites and only include secondary and tertiary dendrites) and (3) having at least 1 spine (to exclude aspinous dendrites from interneurons). PSD area was calculated by multiplying the two-dimensional length on each section by average section thickness and the total number of sections on which the PSD appears.

## Immuno-labeling electron microscopy

For immuno-EM analysis of mouse V1, 3 P14 WT mice on a 129/Sve background were transcardially perfused with warm (37°C) 4% PFA, 0.2% glutaraldehyde, 2 mM $CaCl_2$, and 4 mM $MgSO_4$ in 0.1 M cacodylate buffer (pH 7.4) under Tribromoethanol anesthesia. 100 µm thick coronal sections per each mouse, which contain the V1 visual cortex (Bregma −2.5−−3.2 mm, Interaural 1.3–0.6 mm [*Franklin and Paxinos, 2001*]) were cut with a vibratome (Leica) and immersed in the perfusion fixative at 4°C.

At the Electron Microscopy Core at Georgia Regents University, slices were rinsed 3 × 5 min in Hepes buffered saline (HBS). Non-specific binding sites were permeablilized and blocked for 30 min with HBS, 10% BSA (Sigma), and 0.025% Triton X-100 (Sigma). Permeabilization solution was replaced with ice-cold guinea pig anti-VGlut1 1:750 (Millipore) in HBS plus 1% BSA and 0.0025% Triton X-100, and slices were incubated at 4°C overnight on a shaker. After washing 3 × 5 min in HBS-0.05% BSA, slices were incubated in anti-guinea pig Nanogold 1:250 (Nanoprobes, NY) at 4°C overnight on a shaker. Slices were washed 3 × 5 min in HBS-0.05% BSA, then four changes of distilled $H_2O$ for 2 hr, then incubated for 2 hr on a shaker in 0.5 ml of GoldEnhance EM (Nanoprobes) mixed according to manufacturer's directions. Slices were washed thoroughly in ice-cold $H_2O$ to stop the gold enhancement. After washing 2 × 5 min in HBS, slices were incubated in rat anti-VGlut2 1:250 (MabTechnologies, Inc., GA) in HBS plus 1% BSA and 0.00025% Triton X-100 at 4°C overnight on a shaker. After washing 3 × 5 min in HBS-0.05% BSA, slices were incubated in anti-rat Nanogold 1:250 (Nanoprobes) at 4°C overnight on a shaker. Slices were washed 3 × 5 min in HBS-0.05% BSA, then fixed at room temperature in perfusion fixative for 20 min. After four changes of distilled $H_2O$ for 2 hr, slices were incubated for 3 hr on a shaker in 0.5 ml of GoldEnhance EM solution. Slices were washed thoroughly in ice-cold $H_2O$ to stop the gold enhancement, washed 2 × 5 min in HBS, then washed 3 × 5 min in 0.1 M cacodylate buffer in preparation for processing and embedding. Slices were post-fixed in 1% OsO4 plus 1/5% potassium ferrocyanide in cacodylate buffer for 1 hr. Slices were washed 3 × 10 min in cacodylate buffer, post-fixed for 1 hr in 1% $OsO_4$ in cacodylate buffer, then washed 3 × 10 min in distilled $H_2O$. Slices were stained in 2% aqueous uranyl acetate on a shaker at room temperature for 1 hr, then washed 3 × 5 min in distilled $H_2O$. They were then dehydrated in an ascending ethanol series (50%, 70%, 90% and 100%) for 5–10 min each, with 100% repeated 3 × 10 min. Slices went through 2 × 10 min changes of propylene oxide, were placed in a 1:1 mixture of propylene oxide: Embed 812 resin mixture (EMS) for 1 hr, then put in 100% Embed 812 overnight on a rotator. Slices were flat embedded so that the plane of sectioning was perpendicular to the slice's surface, polymerized at 60°C for 24 hr.

Thin sections were cut with a diamond knife on a Leica EM UC6 ultramicrotome, collected on copper grids and stained with lead citrate. Sections were observed in a JEM 1230 transmission electron microscope (JEOL, Japan) at 110 kV. Areas positioned 50–100 μm below the pial surface corresponding to the synaptic zone (a.k.a. layer I), containing the dendrites of layer II/III neurons, were imaged with an UltraScan 4000 CCD camera and First Light Digital Camera Controller (Gatan Inc., PA).

## Structured illumination microscopy (SIM)

WT brains were harvested and cryoprotected at P15 following 4% PFA fixation. Sections (20 μm) were cut on a cryostat (Leica) and stained for IHC using primary antibodies against VGlut1 (1:500; guinea pig; Millipore) and VGlut2 (1:750; rabbit; Synaptic Systems) followed by Alexa-conjugated secondary antibodies. Sections were imaged using a Zeiss ELYRA PS1 microscope. 3D structured illumination images of the S/Z of V1 were captured and images subsequently processed using Zeiss SIM algorithms.

To quantify co-localized VGlut1 and VGlut2 puncta, SIM-processed image files were opened in Imaris (Bitplane, Switzerland) and spot channels generated for the synaptic markers using dimensions determined empirically from averaged measurements. Matlab (Mathworks, MA) was subsequently used to only show those puncta within 100 nm, 200 nm, or 300 nm of one another.

## In utero electroporation (IUE) and 3D analysis of confocally-imaged synaptic structures

Timed pregnant wild type WT (CD1, Charles River, MA) and hevin KO (129/Sve) dams were utilized for IUE. All electroporations were performed at embryonic day (E) 15.5 in order to target neocortical layer 2/3 pyramidal neurons. Dams were sedated with continuously vaporized isofluorane and cesarean sectioned to expose both uterine horns. 1 μg of DNA plasmid containing shControl-CAG-EGFP with loading dye was injected into one lateral ventricle of each embryo using a pulled glass pipette. Five 50 ms pulses of 50 V spaced 950 ms apart were applied with tweezertrodes (positive paddle against the skull over the injection site, the negative paddle across the body away from the placenta) using the BTX ECM 830 (Harvard Apparatus, MA). Warm PBS was applied to embryos and dam to prevent drying. Following the electroporation, the uterine horns were returned to the abdominal cavity and the peritoneum, anterior muscle, and skin were sutured separately. The dam was then placed on a heating pad to recover and monitored daily following the surgery. All procedures for animal surgery and maintenance were performed in accordance with Duke Institutional Animal Care and Use Committee.

Electroporated brains were harvested and cryoprotected at P21 following 4% PFA fixation. Sections (40 μm) were cut on a cryostat (Leica) and stained for IHC using primary antibodies against GFP (1:750; chicken; Millipore), VGlut1 (1:500; guinea pig; Millipore), and VGlut2 (1:750; rabbit; Synaptic Systems) followed by Alexa-conjugated secondary antibodies. GFP-expressing secondary/tertiary dendrites in the S/Z, along with surrounding VGlut1/2 presynaptic puncta, were imaged on a Zeiss 780 inverted confocal microscope at 63× with 8× optical zoom at 0.13 μm optical section thickness. Z-stacks were deconvolved with Huygens image processing software (Scientific Volume Imaging, The Netherlands) and then imported into Imaris for analysis.

In Imaris, dendrites were reconstructed in 3D using either the Surfaces or FilamentTracer tool. Discrete VGlut puncta were resolved with the Spots tool. Presynaptic puncta within 0.2 μm of dendrites were then isolated using the Find Spots Close to Filament/Surface Matlab algorithm. Spines were then quantified by eye on the basis of their associated presynaptic partners, with the analyst blinded as to the genotype.

## Statistics

Statistica (StatSoft, OK) was used for all statistical analyses. Variability between different IHC synaptic staining pairs was controlled for by the use of nested design hierarchical ANOVAs (under Generalized Linear Models in Statistica) with experimental pair nested within condition. Graphical data are presented as mean ± s.e.m.

## Acknowledgements

We would like to thank Sara Miller and Neil Medvitz (Duke University Electron Microscopy Service) as well as Libby Perry and Brendan Marshall (Electron Microscopy Core at Georgia Regents University) for

their excellent technical assistance, Rebecca Klein (Department of Psychiatry and Behavioral Sciences, Duke University) for assistance with Golgi-cox imaging and analysis, and Ben Philpot (University of North Carolina School of Medicine) and lab members for technical advising on cortical slice electrophysiology methods. This research was funded by NIH/NIDA R01 DA031833 to CE, training grant 2T32NS51156-6 and NRSA 1F32NS083283-01A1 to WCR, R01s NS059957 and MH103374 to SS, R01 NS083897 to DLS, R01 NS071008 to BS, and the Holland-Trice Fellowship, Wakeman Fellowship and T32 GM007171-Medical Scientist Training Program to SB CE was an Esther and Joseph Klingenstein Fund Fellow and Alfred P Sloan Fellow.

## Additional information

### Funding

| Funder | Grant reference number | Author |
|---|---|---|
| National Institute on Drug Abuse | R01 DA031833 | Cagla Eroglu |
| Esther A. and Joseph Klingenstein Fund | Fellowship | Cagla Eroglu |
| Alfred P. Sloan Foundation | Fellowship | Cagla Eroglu |
| National Institute of Neurological Disorders and Stroke | 1F32NS083283 | W Christopher Risher |
| National Institutes of Health | 2T32NS51156-6 | W Christopher Risher |
| National Institute of Neurological Disorders and Stroke | NS059957 | Scott H Soderling |
| National Institute of Mental Health | MH103374 | Scott H Soderling |
| National Institute of Neurological Disorders and Stroke | NS083897 | Debra L Silver |
| Duke University School of Medicine | Holland-Trice Fellowship | Srishti Bhagat |
| Duke University School of Medicine | Wakeman Fellowship | Srishti Bhagat |
| National Institutes of Health | T32 GM007171-Medical Scientist Training Grant | Srishti Bhagat |
| National Institute of Neurological Disorders and Stroke | NS071008 | Beth Stevens |

The funders had no role in study design, data collection and interpretation, or the decision to submit the work for publication.

### Author contributions

WCR, Conception and design, Acquisition of data, Analysis and interpretation of data, Drafting or revising the article, Contributed unpublished essential data or reagents; SP, IHK, AU, DKW, L-JP, JSA, OYC, Acquisition of data, Analysis and interpretation of data; SB, Conception and design, Acquisition of data, Analysis and interpretation of data; DLS, BS, Analysis and interpretation of data, Drafting or revising the article; NC, SHS, Conception and design, Analysis and interpretation of data, Drafting or revising the article; CE, Conception and design, Acquisition of data, Analysis and interpretation of data, Drafting or revising the article

### Author ORCIDs

W Christopher Risher, http://orcid.org/0000-0002-2230-2865
Cagla Eroglu, http://orcid.org/0000-0002-7204-0218

### Ethics

Animal experimentation: This study was performed in strict accordance with the recommendations in the Guide for the Care and Use of Laboratory Animals of the National Institutes of Health. All of the animals were handled according to approved institutional animal care and use committee (IACUC) protocol (# A195-11-08) of Duke University Medical Center. The mice were euthanized by following the approved protocols which were performed under avertin anesthesia, and every effort was made to minimize suffering.

## Additional files

**Supplementary file**

• Source code 1. Puncta Analyzer plugin for ImageJ. This plugin allows for the quantification of co-localized synaptic puncta from immuno-stained cells or tissue. The plugin separates the original image file into red and green channels, subtracts background (rolling ball radius = 50), and asks the user to threshold each channel individually in order to detect discrete puncta without introducing noise. The Puncta Analyzer plugin then uses an algorithm to detect the number of puncta that are in close alignment across the two channels, yielding quantified co-localized puncta.

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
