## [Decision Letter]

Thank you for sending your work entitled “Astrocytes refine cortical connectivity by regulating synaptic competition at dendritic spines” for consideration at *eLife*. Your article has been favorably evaluated by Eve Marder (Senior editor) and 3 reviewers, one of whom is a member of our Board of Reviewing Editors.

The Reviewing editor and the other reviewers discussed their comments before we reached this decision, and the Reviewing editor has assembled the following comments to help you prepare a revised submission.

The reviewers are enthusiastic about your interesting findings that spines with multiple excitatory inputs (SMECs) derive from intracortical and thalamic inputs formed during development and resolved during maturation, and that astrocyte-secreted Hevin plays an important role in promoting thalamocortical synapses. The reviewers are also impressed by the intensive EM reconstruction you used to document these findings. On the other hand, the reviewers also feel that the evidence for competition is correlative and weak at this point, and the mechanisms by which Hevin bias thalamocortical vs. intracortical synapse formation is unclear. On balance, we invite you to submit a revised manuscript addressing the following specific critiques:

1) The immuno–EM studies constitute a key piece of evidence to support that SMECs contain both vGluT2+ thalamocortical and vGluT1+ intracortical synapses. However, the presentation is entirely anecdotal. More quantitative description is required.

2) Given the concern expressed in point #3 below and based on the evidence presented, one cannot rule out a scenario of delayed maturation in the Hevin KO. Perhaps growth and innervation are slower in general, but since intracortical synapse formation is earlier, at the P25 time point it has already 'caught up' but the thalamic axons are not all there yet, and synaptic competition is also delayed. You need to provide evidence against this synaptic delay alternative explanation.

3) The following critiques are related specifically to the conclusions regarding synaptic competition: while this is an interesting possibility, the reviewers feel that current data do not provide sufficient evidence to justify putting competition in the Title, Abstract, and as a major conclusion of the paper. While *eLife* has a policy to not require extensive new experiments, the following experiments, while not necessarily required for acceptance, would, in our opinion, greatly strengthen the paper. We hope that you can address as many of these issues as possible. We hope that you may have some relevant data in hand, and remind you that *eLife* has no restriction on the number of figures. Of course, brevity is always a virtue, but if you have data that was omitted because of length restrictions at most journals, we would rather have the full story. Otherwise the rhetoric of the paper needs to be significantly toned down, and it needs to focus more specifically on the findings for which you have strong data.

3a) The conclusions regarding synaptic competition between intra-cortical and thalamo-cortical inputs are based on several points of evidence, none of which are particularly strong. The main evidence is increased in cortical VGlut1 and decreased VGlut2 staining at P25. Although you show in Figure 1–figure supplement 3 that the number of LGN neurons is normal in the Hevin KO, you do not show (despite the misleading title of this figure) that thalamic ingrowth into cortex and thalamic axonal arborization is intact. You need to show that thalamic axons needed for synapse formation in the cortex are intact in the Hevin KO, and that the fewer VGlut2 synapses are not due to deficiencies such as reduced thalamic ingrowth or arborization. The fact that immunohistochemistry from P7 mice already shows reduced VGlut2 staining at a time point where SMECs are expected to be similar between WT and KO mice, and Hevin levels are relatively low strengthens this concern. You could further bolster the claim of synaptic competition by demonstrating that cortical expression of Hevin KO rescues the VGlut2 staining loss in cortex.

3b) A second point of evidence that you present for reduced synaptic competition at SMECs between thalamo-cortical and intra-cortical synapses in the Hevin KO, is that, while the apparent overlap of VGlut1 and VGlut2 staining is reduced in WT mice from P15 to P25, in the Hevin KO overlap at P25 is equivalent to P15 WT. The caveat raised above applies here too. One would also like to see the P15 Hevin KO data in this comparison. Similar WT and KO overlap at P15 would support your hypothesis. It would not if there is already reduced overlap at P15 (which one might imagine if initial ingrowth or axon elaboration is deficient).

3c) The third and weakest point of evidence is the immuno–EM showing that some normal SMECs at P14 are dually innervated by VGlut1 and VGlut2 positive terminals. While interesting, these data are anecdotal (see major concern #1). There is no quantification of % SMECs innervated by both VGlut1 and VGlut2 at P14 vs P25 as opposed to only VGlut1 (or VGlut2) synapses. The reduction in SMECs with age has already been reported, so the fact that they represent sites of synaptic competition during development is not new. The main added value would have been to show that this competition is relevant to a significant % of thalamic vs intracortical inputs, and that it is this particular aspect of the competition that is deficient in the Hevin KO. In other words, since thalamic axons are fewer in general, one would expect that at P15, when the WT and KO have the same number of SMECs in both cases, most of these would represent multiple VGlut1 contacts. With maturation each WT spine would retain only a single synapse, VGlut1 OR VGlut2. In the case of the KO one should also see a reduction in SMECs, but they should be biased towards VGlut1 contacts. The fact that SMECs are still so abundant in the P25 KO suggests a general problem with maturation/competition rather than a thalamus specific problem.

3d) How does the dendritic vs spine targeting by incoming afferents fit with your general thalamic vs cortical competition scenario? It could also be an issue of maturation, or maybe just a targeting problem in the KO?

---

## [Author Response]

*1) The immuno–EM studies constitute a key piece of evidence to support that SMECs contain both VGluT2+ thalamocortical and VGluT1+ intracortical synapses. However, the presentation is entirely anecdotal. More quantitative description is required*.

We agree with the reviewers that the immuno–EM finding, as described in the original manuscript, was anecdotal. We had initially intended to provide quantification for this analysis, but we were faced with a major technical bottleneck. To visualize SMECs in 2D electron micrographs, even at P15 when they are abundant, is extremely difficult due to their unique geometry. Only 3D reconstructions obtained from serial sections can properly resolve SMECs at an appreciable rate in EM. After careful observation of over 200 two-dimensional electron micrographs of VGlut1/VGlut2 immuno-labeled tissue, we were only able to capture one single SMEC. This SMEC was innervated by VGlut1- and VGlut2-positive terminals (which is the image that appears in the manuscript).

Immuno-labeling in EM requires certain tissue permeation techniques, which are not compatible with serial sectioning. Therefore, the merger of these two techniques at EM-level represented a steep technical challenge that was not possible to complete within the time frame allotted for revisions. However, in the revised manuscript we devised a different method to address this important concern raised by the reviewers. To do so, we labeled dendrites from layer 2/3 cortical pyramidal neurons in WT and Hevin KO mice using in utero electroporation (IUE) at embryonic day 15.5 to sparsely transfect them with a plasmid that expresses GFP. These IUEs were performed in collaboration with our colleague Dr. Debra Silver’s lab. The brains were harvested postnatally and stained for VGlut1 and VGlut2 to mark presynaptic terminals. The secondary and tertiary dendrites of the sparsely GFP-labeled Layer 2/3 neurons were imaged with a confocal microscope. Confocal Z-stacks were then imported into Huygens image processing software for deconvolution. Dendrites, spines, and VGlut1/VGlut2 puncta were then reconstructed in Imaris in 3D. Using a “Find spots close to surfaces” Matlab algorithm, we could isolate the presynaptic puncta that were in close proximity (0.2 µm) to the dendrite. This procedure allowed us to image potential SMECs in a quantitative fashion. We first verified this technique on P15 WT samples and found that this method yields SMEC counts that are similar to our results with 3D ssEM. Then we performed the same analyses on P21 Hevin KO and WT mice (blind to genotype) to study the effect of lack of hevin on the composition of the inputs onto SMECs. We found that the majority of the SMECs are doubly innervated by VGlut1/VGlut2 positive terminals. Interestingly, in line with our 3D EM data, we found that the abundance of SMECs and particularly SMECs with simultaneous VGlut1/2 partners was significantly increased in Hevin KO mice. Taken together, these results, which are presented in Figure 8 and Figure 8—figure supplement 2 of the revised manuscript, strengthen our findings that SMECs are primarily sites for simultaneous contact for intracortical and thalamocortical axons. Our results also indicate that in Hevin KO mice, the VGlut2+ thalamic inputs cannot stabilize their individual spine contacts, thus they continue to be associated with VGlut1+ intracortical inputs at SMECs; this proposed model is presented in Figure 8.

*2) Given the concern expressed in point #3 below and based on the evidence presented, one cannot rule out a scenario of delayed maturation in the Hevin KO. Perhaps growth and innervation are slower in general, but since intracortical synapse formation is earlier, at the P25 time point it has already 'caught up' but the thalamic axons are not all there yet, and synaptic competition is also delayed. You need to provide evidence against this synaptic delay alternative explanation*.

In this revised manuscript we included four new experiments that address the concern raised regarding the possibility that thalamocortical synapses are reduced in Hevin KOs due to a developmental delay in thalamic axonal innervation and/or thalamocortical synapse formation. We believe our new findings provide strong evidence that hevin is specifically required for formation and maintenance of thalamocortical synapses.

We also would like to clarify that intracortical synapse formation, especially at the synaptic zone of the cortex, is not thought to be an earlier event then thalamocortical synaptogenesis. During cortical development, most of the intracortical and thalamocortical synapses form in parallel during the peak synaptogenic window (P5-P21) (39; 34); this occurs after thalamic axons reach their target regions in cortex, an event that is finalized within the first few days after birth (P1-P4) (Sur and Leamey, 2001; [13]). In our revised manuscript, we now include a detailed Introduction that outlines the current knowledge on cortical synapse formation.

Experiment 1: We agree that the loss of thalamocortical synapses in Hevin KOs could be due to a problem with thalamic axons. To determine if, in Hevin KOs, the thalamocortical axons efficiently innervated the cortex, in particular the synaptic zone (S/Z, layer 1) of V1, we used a Flex-AAV-GFP viral delivery system in combination with rabies virus glycoprotein-coated Lenti-FuGB2-Cre. This way we were able to specifically trace the thalamic axons that originate from the dorsal lateral geniculate nucleus (dLGN) of thalamus innervating the V1 (Figure 2—figure supplement 3). These experiments were performed in collaboration with our colleague Dr. Scott Soderling’s lab. This method allows us to observe long-distance projections whose source neurons and target regions are precisely known. Our results showed that thalamic axons in the Hevin KOs had no difficulty reaching the synaptic zone during the period of thalamocortical synapse formation and maturation. The number of labeled neurons varies from animal to animal due to the reliance on viral infection at two distinct sites, which precludes quantification in this type of analysis. Regardless, we found equal innervation of the S/Z by thalamic axons in Hevin KOs. These results strongly indicate that the defect in thalamocortical synapse numbers we observed in Hevin KOs was not due to a problem with path finding and cortical innervation of thalamic axons.

Experiment 2: The reviewers suggested that a delay in thalamocortical synaptogenesis might provide an alternative explanation for our observations in P25 Hevin KOs. If the decrease we observed in thalamocortical synapses in Hevin KOs at P25 is due to a delay in thalamocortical synaptogenesis, then this delay should later be corrected. To test this, we quantified the number of VGlut/PSD95 synapses in adult (12-week-old) mice. In contrast to the idea that reduced thalamocortical synapses we observed in P25 Hevin KOs is a developmental delay, we found a significant lack of VGlut2+ synapses in adult Hevin KOs compared to WTs (Figure 3). This result indicates that loss of hevin impairs formation and maintenance of thalamocortical synapses. Interestingly, the number of VGlut1+ synapses in adult Hevin KOs was similar to that of the WT adult mice (Figure 3) indicating that the increase we see in the number of VGlut1+ excitatory synapses in Hevin KOs at P7 and P25 is a transient developmental phenomenon.

This transient increase in VGlut1+ cortical synapses may be due to a homeostatic mechanism that compensates for lost thalamic connections, driven by a transient competitive advantage for cortical axons over the thalamic inputs to establish synapses. In line with synaptic compensation, the electrophysiology experiments, that we performed in collaboration with our Duke University colleague Nicole Calakos’s lab on P25 Hevin KO and WT mice, revealed that, despite the severe loss of thalamocortical connections in Hevin KOs, the miniature excitatory postsynaptic currents are normalized to WT levels (new Figure 2—figure supplement 4).

Taken together with our previous findings, these new experiments reveal that hevin is required for normal thalamocortical excitatory synapse formation. In addition, we performed two other experiments to test the sufficiency of hevin to induce thalamocortical synapse formation.

Experiment 3: To determine if hevin was sufficient to induce synapse formation between thalamic axons and cortical neurons, we used a purified neuron culture system. We have utilized neuronal surface antigen L1-CAM specific antibodies to purify cortical and thalamic neurons. In these pure neuronal cultures we tested the ability of hevin to induce formation of VGlut1+ intracortical or VGlut2+ thalamocortical synapses. We found that hevin can significantly induce VGlut2+ positive thalamocortical synapse formation, however hevin does not induce formation of VGlut1+ intracortical synapses. These results are now reported in Figure 4. These in vitro findings show that hevin is sufficient to induce thalamocortical synapse formation in purified neuronal cultures. It is important to note that addition of astrocyte-feeder layers to our purified cortical neurons dramatically stimulates formation of VGlut1+ intracortical synapses (Risher and Eroglu, unpublished results), indicating that there are other astrocyte-secreted factors that induce intracortical synapse formation.

Experiment 4: To determine whether hevin is sufficient to induce thalamocortical synapse formation in vivo, we injected purified hevin protein into the V1 cortices of Hevin KOs and determined whether hevin injection can induce VGlut2+ thalamocortical synapse formation. In our optimization experiments, we determined that injected hevin persists up to 3 days in the cortex. Therefore, we harvested the brains at 3 days post-hevin injection and determined the synapse composition close to the sites of injection. Vehicle-injected littermates served as controls. We found that hevin injection leads to a profound increase in VGlut2+ synapse formation in the cortex, whereas hevin injection does not alter VGlut1+ synapse numbers. These data are now presented in Figure 4 and Figure 4—figure supplement 1. Taken together, these in vitro and in vivo experiments demonstrate that hevin is sufficient to induce thalamocortical synapse formation.

*3) The following critiques are related specifically to the conclusions regarding synaptic competition: while this is an interesting possibility, the reviewers feel that current data do not provide sufficient evidence to justify putting competition in the Title, Abstract, and as a major conclusion of the paper. While eLife has a policy to not require extensive new experiments, the following experiments, while not necessarily required for acceptance, would, in our opinion, greatly strengthen the paper. We hope that you can address as many of these issues as possible. We hope that you may have some relevant data in hand, and remind you that eLife has no restriction on the number of figures. Of course, brevity is always a virtue, but if you have data that was omitted because of length restrictions at most journals, we would rather have the full story. Otherwise the rhetoric of the paper needs to be significantly toned down, and it needs to focus more specifically on the findings for which you have strong data*.

We trust that with the addition of above-mentioned experiments, we greatly strengthened our manuscript and addressed the majority, if not all the points, which were raised within the third concern. We believe our manuscript now makes a compelling case for the SMECs as potential sites for competition between thalamic and cortical axons. However, we agree that further experiments are required to decisively show the role of these structures in synaptic competition. Therefore, in line with the reviewers’ suggestions we removed competition from the Title and the Abstract as a major conclusion of the paper, but left it in as an interesting suggestion that can be drawn from our findings.

*3a) The conclusions regarding synaptic competition between intra-cortical and thalamo-cortical inputs are based on several points of evidence, none of which are particularly strong. The main evidence is increased in cortical VGlut1 and decreased VGlut2 staining at P25. Although you show in Figure 1–figure supplement 3 that the number of LGN neurons is normal in the Hevin KO, you do not show (despite the misleading title of this figure) that thalamic ingrowth into cortex and thalamic axonal arborization is intact. You need to show that thalamic axons needed for synapse formation in the cortex are intact in the Hevin KO, and that the fewer VGlut2 synapses are not due to deficiencies such as reduced thalamic ingrowth or arborization. The fact that immunohistochemistry from P7 mice already shows reduced VGlut2 staining at a time point where SMECs are expected to be similar between WT and KO mice, and Hevin levels are relatively low strengthens this concern. You could further bolster the claim of synaptic competition by demonstrating that cortical expression of Hevin KO rescues the VGlut2 staining loss in cortex*.

Our revised manuscript includes data that address these important points and shows that lack of VGlut2+ thalamic inputs in the Hevin KOs is not due to a problem with the innervation of thalamic axons. We kindly refer the reviewers to our response to major concern #2. In particular, refer to sections with Experiment 1 which demonstrates that thalamic axons needed for synapse formation in the cortex are intact in the Hevin KO and Experiment 4 showing that cortical injection of pure hevin into Hevin KOs stimulates a robust increase in thalamocortical synapses, indicating that cortical expression of hevin can rescue the VGlut2+ terminal loss in KO cortex.

*3b) A second point of evidence that you present for reduced synaptic competition at SMECs between thalamo-cortical and intra-cortical synapses in the Hevin KO, is that, while the apparent overlap of VGlut1 and VGlut2 staining is reduced in WT mice from P15 to P25, in the Hevin KO overlap at P25 is equivalent to P15 WT. The caveat raised above applies here too. One would also like to see the P15 Hevin KO data in this comparison. Similar WT and KO overlap at P15 would support your hypothesis. It would not if there is already reduced overlap at P15 (which one might imagine if initial ingrowth or axon elaboration is deficient)*.

We would like to clarify that we do not claim there is reduced synaptic competition in the Hevin KOs. If that were the case, we would not expect to see SMECs in Hevin KOs. The fact that Hevin KOs still have SMECs at later ages indicates that the “tug-of-war” between thalamic and cortical axons for spines cannot be resolved in these KOs. We propose in our model that, because hevin is required for formation/stabilization of thalamocortical synapses, hevin is also critical for the ability of thalamic axons to withstand the steep competition from the more abundant intracortical terminals at spines. Thus, in Hevin KOs, synaptic competition is rigged towards the stabilization of cortical connections. This model would predict that in Hevin KOs, competition still goes on at thalamocortical contacts at P25 and there are increased numbers of intracortical contacts.

In our original submission, we provided a proxy for the quantification of VGlut1/2-positive SMECs at the confocal microscopy level by reporting the apparent overlap between VGlut1/2 terminals. Our reasoning was that VGlut1/2-positive SMECs might appear as sites of overlap between these two presynaptic markers at the resolution of conventional confocal microscopy. In the revised manuscript, we have included new data investigating the nature of this VGlut1/VGlut2 overlap through the use of high-resolution structured illumination microscopy (SIM). These experiments were done in collaboration with Dr. Beth Stevens’ lab at Boston Children’s Hospital. The lateral resolution of SIM is at least 2-fold greater than confocal microscopy, which allowed us to show that the apparent overlap of VGlut1 and 2 in P15 WT cortex with confocal could essentially be resolved to discrete VGlut1 and VGlut2 puncta with SIM (Figure 8—figure supplement 1). This finding provides strong evidence for overlapping VGlut puncta being the result of closely positioned VGlut1 and VGlut2 puncta rather than the presence of these two markers at the same terminals. These closely associated puncta could potentially be SMECs. However, we now have a better analysis method for SMEC contacts (see major concern #1 and Figure 8), therefore we moved the VGlut1/2 overlap data to Figure 8—figure supplement 1. In line with the reviewers’ suggestion in the revised manuscript, we also analyzed VGlut1/VGlut2 overlap in P15 Hevin KOs (Figure 8—figure supplement 1). Interestingly, the P15 Hevin KOs have an intermediate VGlut1/VGlut2 overlap at P15, which is midway between the P15 WT and P25 WT overlap; this P15 Hevin KO overlap is essentially identical to the elevated overlap levels observed in P25 Hevin KOs. Taken together, these findings show that hevin is required for effective overlap and resolution of the closely associated VGlut1/2 terminals in the cortex. Provided that the overlap we see between VGlut1 and 2 are representative of SMECs, the fact that Hevin KOs have an intermediate phenotype both at P15 and P25 suggests that, in the absence of hevin, both the efficient formation and resolution of VGlut1- and 2-innervated SMECs is impaired. We also tested this point in the revised manuscript (we kindly refer the reviewers to responses #1 and #3c).

*3c) The third and weakest point of evidence is the immuno–EM showing that some normal SMECs at P14 are dually innervated by VGlut1 and VGlut2 positive terminals. While interesting, these data are anecdotal (see major concern #1). There is no quantification of % SMECs innervated by both VGlut1 and VGlut2 at P14 vs P25 as opposed to only VGlut1 (or VGlut2) synapses. The reduction in SMECs with age has already been reported, so the fact that they represent sites of synaptic competition during development is not new. The main added value would have been to show that this competition is relevant to a significant % of thalamic vs intracortical inputs, and that it is this particular aspect of the competition that is deficient in the Hevin KO. In other words, since thalamic axons are fewer in general, one would expect that at P15, when the WT and KO have the same number of SMECs in both cases, most of these would represent multiple VGlut1 contacts. With maturation each WT spine would retain only a single synapse, VGlut1 OR VGlut2. In the case of the KO one should also see a reduction in SMECs, but they should be biased towards VGlut1 contacts. The fact that SMECs are still so abundant in the P25 KO suggests a general problem with maturation/competition rather than a thalamus specific problem*.

We refer the reviewers to major concern #1 with regards to the SMEC input quantification. We respectfully disagree with the reviewers’ comment: “the reduction in SMECs with age has already been reported, so the fact that they represent sites of synaptic competition during development is not new”. To our knowledge, this is the first study to show that SMECs are developmentally regulated in the cortex, and that they can be innervated by different populations of axons simultaneously. In addition, the revised manuscript includes quantitative evidence to support the idea of SMECs as sites for simultaneous contact between thalamic and cortical axons. As pointed out by the reviewers, the overall loss of VGlut2+ terminals in the Hevin KOs would suggest that remaining SMECs in Hevin KOs would more likely be innervated by multiple VGlut1+ contacts. However, our Imaris-assisted 3D reconstructions of GFP-filled dendrites showed that the majority of SMECs were in fact of the mixed VGlut1/VGlut2 subtype (Figure 8). This particular enrichment of VGlut1/2-positive SMECs indicates that these structures are sites of competition mainly between thalamic and cortical inputs. We propose a model for hevin’s regulation of thalamocortical connectivity at dendritic spines (Figure 8) in accordance with these findings. Future studies investigating the mechanism driving the close association of cortical and thalamic inputs on dendrites are needed to fully understand the purpose of these structures, but as the reviewers would agree this would be beyond the scope of this study.

3d) How does the dendritic vs spine targeting by incoming afferents fit with your general thalamic vs cortical competition scenario? It could also be an issue of maturation, or maybe just a targeting problem in the KO?

The enrichment of shaft synapses in Hevin KOs is indeed an interesting observation and may be a phenomenon that is distinct from the competition occurring at SMECs. Shaft synapses have been described in cortical dendrites, particularly during early stages of synaptic development, and they may be an intermediate stage before the formation of spine synapses. The persistence of shaft synapses in P25 Hevin KOs thus may be a part of the overall immaturity of Hevin KO dendrites, indicating that hevin and/or establishment of the proper thalamic input is necessary for the maturation of dendrites. We now include a part in the Discussion on our findings on shaft synapses.